



**On the Influence of Spatial Heterogeneity of Runoff Generation on the**
**Distributed Unit Hydrograph for Flood Prediction**
Bin Yi[1,2], Lu Chen[1,2,3]*, Tao Xie[1,2]
1. *School of Civil and Hydraulic Engineering, Huazhong University of Science and Technology, Wuhan, 430074, China*
2. *Hubei Key Laboratory of Digital Valley Science and Technology, Wuhan 430074, China*
3. *School of Water Resources and Civil Engineering, Tibet Agricultural & Animal Husbandry University, Linzhi 860000,*
*China*
Correspondence: Lu Chen (chen_lu@hust.edu.cn)
**Abstract:** The spatial scale mismatch between runoff generation and runoff routing is an acceptable
compromise but a common issue in challenging hydrological modelling. Moreover, there is hardly any
report available on whether unit hydrograph (UH) that is consistent with the spatial scale of runoff
generation can be computed. The objective of this study was to explore the influence of spatial
heterogeneity of runoff generation on the UH for flood prediction. To this end, a novel GIS-based
dynamic time-varying unit hydrograph (DTDUH) was proposed based on the time-varying unit
hydrograph (TDUH). The DTDUH can be defined as a typical hydrograph of direct runoff which gets
generated from one centimeter of effective rainfall falling at a uniform rate over the saturated drainage
basin uniformly during a specific duration. The DTDUH was computed based on the saturated areas
of the watershed instead of the global watershed. The saturated areas were extracted based on the TWI.
Finally, the Longhu River basin and Dongshi River basin were selected as two case studies. Results
showed that the proposed method exhibited consistent or better performance compared with that of the
linear reservoir routing method, and performed better than the TDUH method. Specifically, the
DTDUH method indicated good performances for the flood events with low antecedent soil moisture,
and it performed consistently with the TDUH when the global watershed is nearly saturated. The



proposed method can be used for the watersheds with sparse gauging stations and limited observed
rainfall and runoff data, as is the same with the TDUH method. Simultaneously, it is well applicable
for the humid or mountain watershed where the saturation-excess rainfall is the dominant.
**Keywords:** Hydrological modelling; Distributed unit hydrograph; Runoff routing; Runoff generation;
XAJ model

## 1. Introduction

The assumption that basins behave as linear systems (i.e., there is proportionality and additivity

between excess rainfall and total storm response) has been the core to hydrology over the past century
(Goodrich et al., 1997; Bunster et al., 2019). On such condition, a single response function named the
UH has been widely applied to acquire the stormflow at the basin outlet (Czyzyk et al., 2020). The UH
can be developed for both gauged and ungauged watersheds (Monajemi et al., 2021). For gauged basins,
unit hydrographs are derived from observed data by measuring rainfall and runoff data. For ungauged
basin, some synthetic methods, such as the Snyder's method (Snyder, 1938) and Gray's method (Gray,
1961), are used to determine the unit hydrographs from spatially averaged basin characteristics. One
variation is the synthetic UH method proposed by Clark (1945), in which the UH was derived by
integrating two basic features of a watershed rainfall-runoff process. Specifically, translation through
water movement was characterized by time-area method, and linear reservoir routing was used to
represent attenuation through storage (Cho et al., 2018; Bunster et al., 2019). This history of
development was synthetized in the works of Rodríguez-Iturbe and Valdés (1979) and Gupta et al.
(1980), who proposed the geomorphological instantaneous unit hydrograph (GIUH). Subsequently, the
width function-based geomorphological instantaneous unit hydrograph (WFIUH) method has been





formulated with the development of digital elevation models (DEMs) and geographic information
system (GIS) technology (Seo et al., 2016). The WFIUH was derived by combining the flow paths to
the outlet given by the width function with the flow celerity along the flow paths (Kirkby, 1976).
The UH method assumes the watershed response to be linear and time invariant, and rainfall to
be spatially homogeneous. In contrast to the linearity assumption, basins have been shown to exhibit
nonlinearity in the transformation of excess rainfall to stormflow (Bunster et al., 2019). For a small
watershed, Minshall (1960) showed that significantly different UHs were produced by different rainfall
intensities. To cope with this nonlinearity, Rodríguez-Iturbe et al. (1982) extended the GIUH to the
geomorphoclimatic IUH (GcIUH) by incorporating excess rainfall intensity. Lee et al. (2008) proposed
a variable kinematic wave GIUH accounting for time-varying rainfall intensity, which may be
applicable to ungauged catchments that are influenced by high intensity rainfall. To this end, it is
necessary to consider the geomorphic characteristics of the watershed and incorporate time-varying
rainfall intensity in the rainfall-runoff modelling processes.
Spatially distributed travel time models, also known as Spatially Distributed Unit Hydrograph
models (DUH) (Maidment et al., 1996), are a semi-analytical form of the WFIUH identified by Rigon
et al. (2016) in which spatially distributed flow celerity associated with the watershed characteristics
and temporally varying excess rainfall rates can be considered. In this method, the travel time of each
grid cell can be calculated by dividing the travel distance of a cell to the next cell by the velocity of
flow generated in that cell (Paul et al., 2018). The travel time is then summed along the flow path to
obtain the total travel time from each cell to the outlet. The DUH is thus derived using the distribution
of travel time from all grid cells in a watershed. Several aspects (i.e. Rainfall intensities, upstream





contributions, watershed equilibrium) were attracted much attention to obtain more accurate travel
time distributions. Some DUH methods assumed a time-invariant travel time field and ignored the
dependence of travel time on excess rainfall intensity (Melesse and Graham, 2004; Noto and La Loggia,
2007; Gibbs et al., 2010; Grimaldi et al., 2010), while others suggested various UHs corresponding to
different storm events, namely time-varying distributed unit hydrograph (TDUH) (Martinez et al.,
2002; Sarangi et al., 2007; Du et al., 2009). The upstream contributions to the travel time estimation
should also been considered in the time-varying DUH method, which was developed from a static
upstream contribution to a dynamic contribution. For example, Bunster et al. (2019) developed a
TDUH model that accounts for dynamic upstream contributions and compared its performance against
other TDUH methods, and characterized the temporal behavior of upstream contributions and its
impact on travel times in the basin. However, the methods discussed above assumed that equilibrium
in each individual grid cell or global watershed can be reached before the end of the rainfall excess
pulse. To this end, Yi et al. (2022) proposed a time-varying distributed unit hydrograph method for
runoff routing that accounts for dynamic rainfall intensity and soil moisture content, namely the time-
varying distributed unit hydrograph considering soil moisture content (TDUH-MC). Hydrologists have
made great efforts to address the nonlinear issues of the UHs in the past decades, while these
approximations are still acceptable compromises in challenging hydrology research.

Nevertheless, we found that these approximations discussed above neglected the influence of

spatial heterogeneity of runoff generation on the UH. For instance, in a humid watershed, the excess
rainfall can be more inclined to happen at near-channel areas, and sometimes in the arid zones with
long-storms duration according to the saturation-excess mechanism (Li and Sivapalan, 2014). In





previous studies, the time-varying rainfall intensities were commonly considered to obtain a more
accurate travel time distributions to compute the UH for the whole basin, while the depth of the excess
rainfall was also considered to be uniformly distributed throughout the whole watershed (the excess
rainfall actually comes from local saturated areas). The outflow hydrograph is then calculated by
superimposing the response to each individual excess rainfall pulse, or equivalently by convoluting
the spatially averaged excess rainfall and a UH obtained from the IUH. This indeed raised a problem
that whether the unit hydrograph can reflect the realities of the runoff routing processes of an actual
watershed.
The problem of spatial scale mismatch between runoff generation and confluence theory is
prevalent in hydrological modeling, and hydrologists almost ignore the forecasting errors associated
with this issue. For example, the Xinanjiang (XAJ) model is a conceptual hydrological model proposed
by Zhao et al. (1980) for flood forecasts in the Xin'an River basin. It has been widely used in humid
and semi-humid watersheds all over the world (Zhao, 1992; Zhou et al., 2019; Huang et al., 2020). In
the model, two parabolic curves are adopted to represent the spatially non-uniform distribution of the
tension water storage and the free water storage. In order to match the spatial scales of the runoff
generation and the runoff routing, the excess rainfall generated in the saturated areas was assumed to
be uniformly distributed across the whole basin. However, this assumption may result in huge errors,
and there is hardly any report available on whether UH that is consistent with the spatial scale of runoff
generation can be computed.
The objective of this study was therefore to explore the influence of spatial heterogeneity of runoff
generation on the UH for flood prediction. The main contributions of the present study are given as



follows. 1) The DUH method was used as the basic tool to compute the DTDUH corresponding to
various saturated states of the watershed. 2) The definition of the DTDUH was different with the
traditional TDUH as it represented the characteristics of the runoff generated areas instead of the whole
basin. 3) The XAJ model was used as the hydrological modelling framework to unify the spatial scales
of the runoff generation and the confluence method. The performance of the DTDUH method was
compared with that of the traditional TDUH method based on flood events. Finally, the Longhu River
basin and the Dongshi River basin in the Guangdong Province, China, were selected as two case studies.
The influence of spatial heterogeneity of runoff generation on the UH for flood prediction was
investigated.

## 2. Hydrological models

### 2.1 Runoff generation based on the saturation-excess mechanism

Saturation-excess runoff is the major runoff mechanism in humid well-vegetated areas where
infiltration rates often exceed rainfall intensity (Tromp-Van Meerveld and McDonnell, 2006; Hoang
et al., 2017). Many hydrological models, such as the TOPMODEL (Beven and Kirkby, 1979), the
Variable Infiltration Capacity (VIC) model (Liang et al., 1994), the Probability Distributed Model
(PDM) (Moore, 2007), the XAJ model (Zhao, 1992), and the Australian Water Balance Model
(Boughton, 2004), simulate saturation-excess runoff by introducing a statistical distribution of tension
water storage capacity using different methods. Simultaneously, a free water capacity distribution
curve is usually used to divide the total runoff into the surface runoff, interflow and groundwater. In
the XAJ model, two parabolic curves are adopted to represent the spatially non-uniform distribution
of the tension water storage and free water storage. The functional relationships of the tension water





storage capacity curve and free water storage capacity are given by

$$\frac{A_{ps}}{A_p} = 1 - \left(1 - \frac{W}{WMM}\right)^B \tag{1}$$


$$\frac{A_f}{A_s} = 1 - \left(1 - \frac{S}{SMM}\right)^{EX} \tag{2}$$


where $A_{ps}$ is the partial pervious area where the tension water storage capacity is less than or equal
to the value $W$, which is the tension water capacity at a point, varying from 0 to a maximum $WMM$;
$A_p$ is the pervious area of the catchment; $B$ is the exponential of distribution of the tension water
capacity; $A_f$ is the area where the free water storage capacity is less than or equal to the value $S$,
varying from 0 to $SMM$; $A_s$ is the runoff generation area; and $EX$ is the exponential of distribution of
the free water storage capacity curve.
Based on Eq. (1), when rainfall exceeds evaporation, the runoff generated in the saturated areas
can be expressed by

$$R = \begin{cases} PE - WM + W_0 + WM\left(1 - \dfrac{PE + AU}{WMM}\right)^{1+B} & PE + AU < WMM \\ PE - WM + W_0 & PE + AU \geq WMM \end{cases} \tag{3}$$


where $R$ is the total runoff (mm); $WM$ is the areal mean tension water capacity (mm); $PE$ is the rainfall
which exceeds evaporation (mm); $W_0$ is the initial areal mean tension water storage (mm); $AU$ is the
vertical coordinate corresponding to $W_0$.
The total runoff $R$, generated in a wet period in accordance with Eq. (3), can be subsequently
separated into three components, including the surface runoff, interflow and groundwater (Zhao, 1992).

$$RS = \begin{cases} FR\left[PE + S - SM + SM\left(1 - \dfrac{PE + BU}{MS}\right)^{1+EX}\right] & PE + BU < SMM \\ FR(PE + S - SM) & PE + BU \geq SMM \end{cases} \tag{4}$$




$$RI = KI \cdot S \cdot FR \tag{5}$$

$$RG = KG \cdot S \cdot FR \tag{6}$$

where $RS$, $RI$, $RG$ represent the depth of the surface runoff, interflow and groundwater respectively
(mm); $FR$, equalling to $R/PE$, is the proportion of the runoff producing area over the whole basin;
$SM$ is the areal mean free water capacity (mm); $BU$ is the vertical coordinate corresponding to $S$ (mm);
and $KI$ and $KG$ are outflow coefficients of the free water storage to interflow and groundwater,
respectively.
*2.2 Derivation of the DTDUH for the surface runoff routing*

The GIS-derived DUH method was employed for the surface runoff routing calculations, which

allowed the velocity to be calculated on a grid cell basis over the watershed. To remove the linearity
assumption, fully distributed models use routing methods which are usually computationally intensive
because they solve the St. Venant equations (Bunster et al., 2019), so they are usually limited to small
basins. Therefore, the DUH method is an alternative method that allows the use of distributed
information in a much more efficient manner. The TDUH method was used for the computation of the
UH, which considered both the time-varying rainfall intensities and the soil moisture. More details can
be found in Yi et al. (2022). The DTDUH proposed in this study are computed based on the TDUH.
The schematic diagram of the DTDUH method is shown in Fig. 1.

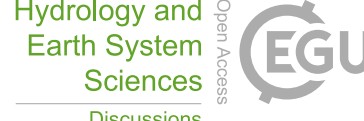


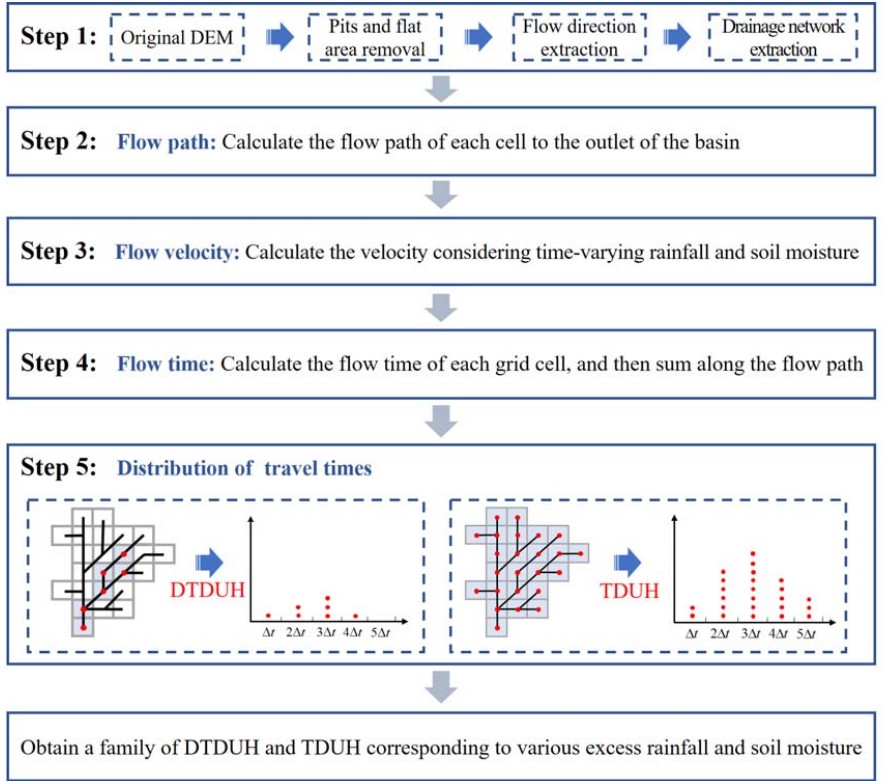


**Figure 1.** Schematic diagram of the DTDUH method. The main differences between the DTDUH and

TDUH lie in that the DTDUH is computed based on local areas while the TDUH is for the global

watershed.

1) The drainage network can be identified based on the advanced DEM preprocessing method

(Grimaldi et al., 2012). Then, the flow path collection ( $\mathbf{Road}_{A_j}$ ) of each gird cell $A_j$ along the flow

directions to the basin outlet can be created. For the whole basin, the flow path collection of all the

grid cells ( $\mathbf{Road}$ ) is expressed by

$$\mathbf{Road} = \left\{ \mathbf{Road}_{A_j} \middle| A_j \in \mathbf{A} \right\} \tag{7}$$

where $A_j$ ( $j = 1, 2, \cdots, N$ ) is the grid cell, $N$ is the total grid cells of the basin; and $\mathbf{A}$ is the collection of

all the grid cell.





The depth of the excess rainfall occurs only in the saturated areas when the entire basin does not
reach a global saturated state, while current methods compute the DUH corresponding to the whole
basin. To this end, we proposed to create a collection of flow paths for the saturated areas, and the
specific formula is given by
$$\mathbf{Road}_{\alpha} = \left\{ \mathbf{Road}_{A_j} \,\middle|\, A_j \in \mathbf{A}_{\alpha} \right\} \tag{8}$$

where  $\mathbf{Road}_{\alpha}$  ( $\mathbf{Road}_{\alpha} \subseteq \mathbf{Road}$ ) is the flow path collection of the saturated grid cells when the soil
moisture proportion is  $\alpha$ ; and  $\mathbf{A}_{\alpha}$  is the collection of the saturated grid cells when the soil moisture
proportion is  $\alpha$ .

2) The flow velocity of each grid cell corresponding to the collection  $\mathbf{Road}_{\alpha}$  is computed based

on the watershed characteristics and the spatial-temporal distribution characteristics of rainfall and soil
moisture, and the specific formula is given as Eq. (9) (Yi et al., 2022).
$$V = k \cdot S^{\frac{1}{2}} \cdot \left( \frac{I_t}{I_c} \right)^{\frac{2}{5}} \cdot \left( \alpha_t \right)^{\gamma} \tag{9}$$

where $V$ (m s$^{-1}$) is the flow velocity; $k$ (m s$^{-1}$) is the land use or flow type coefficient, $S$ (m m$^{-1}$) is the
slope of the grid cell; $I_t$ (mm h$^{-1}$) is the excess rainfall intensity at time $t$; $I_c$ is the reference excess
rainfall intensity of the basin; $\alpha_t$ (unitless) represents the soil moisture content of the basin at time $t$;
and $\gamma$ (unitless) is an exponent smaller than unity, which represents the nonlinear relationship
between soil moisture content and flow velocity.

3) The travel time for each grid cell in collection  $\mathbf{Road}_{\alpha}$  can be calculated by Eq. (10). To

compute the total travel time $\tau_i$ of flow from each cell $i$ to the outlet, travel times along the $R_i$ cells
belonging to the flow path that starts at that cell are added based on Eq. (11).





$$\Delta\tau_j = \frac{L_j}{V} \quad \text{or} \quad \Delta\tau_j = \frac{\sqrt{2}L_j}{V} \tag{10}$$

$$\tau_j = \sum_{A_j \in \mathbf{A}_\alpha} \Delta\tau_j \tag{11}$$

where $\Delta\tau_j$ is the retention time in grid cell $A_j$; $\tau_j$ is the total travel time along the flow path for grid

cell $A_j$; $L_j$ is the grid cell size. When the rasterized flow is flowing along the edges of the grid cell,

the travel length is the cell size $L_j$, whereas the travel length is $\sqrt{2}L_j$ when it is flowing diagonally.

4) Develop a cumulative travel time map of the saturated areas instead of the whole basin based

on cell-by-cell estimates for hillslope velocities. The cumulative travel time map is further divided into

isochrones, which can be used to generate a time-area curve and the resulting DTDUH corresponding

to the collection $\mathbf{Road}_\alpha$ instead of $\mathbf{Road}$.

*2.3 Hydrological modelling framework*

The XAJ model proposed by Zhao et al. (1980) was used as the hydrological modelling

framework. It has been widely used in humid and semi-humid watersheds all over the world. There

are four modules in the model including the evapotranspiration module, runoff generation module,

runoff partition module and runoff routing module. For the evapotranspiration module, the soil profile

of each sub-basin is divided into three layers, the upper, lower and deeper layers, and only when water

in the layer above it has been exhausted does evaporation from the next layer occur. For the runoff

generation and runoff partition modules in the XAJ model, they have been reviewed in Section 2.1.

Finally, for the runoff routing module, subsurface stormflow and subsurface runoff were considered

using a free reservoir. To compare the differences of spatial scale mismatch between the runoff

generation and the runoff routing, linear reservoir, TDUH and DTDUH were selected as the surface

runoff routing methods. The Muskingum method was employed to produce streamflow from channel





to the outlet of the entire basin. The XAJ modelling framework used in this study is given in Fig. 2.

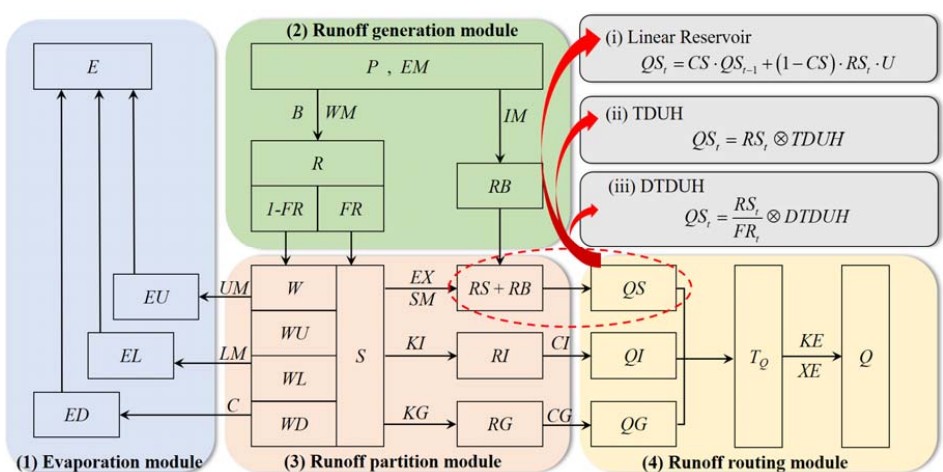


**Figure 2.** Schematic diagram of the XAJ model.

The depths of the surface runoff calculated by Eq. (4) are obtained after a redistribution to the

whole basin. As a matter of fact, the depths of the surface runoff should be calculated only over the
saturated areas when the DTDUH was selected as the surface runoff routing method, which can be
expressed by
$$RS_s^{'} = \frac{RS_s}{FR} \qquad (12)$$

where $RS_s^{'}$ is the depth of the surface runoff corresponding to the saturated areas.

When the linear reservoir, TDUH and DTDUH were used respectively for the surface runoff

routing, the flow discharge at the outlet of the watershed can be computed by
$$QS_t = CS \cdot QS_{t-1} + (1 - CS) \cdot RS_t \cdot \frac{F}{3.6\Delta t} \qquad (13)$$

$$QS_t = RS_t \otimes TDUH \qquad (14)$$

$$QS_t = RS_t^{'} \otimes DTDUH \qquad (15)$$

where $CS$ is the recession constant of the surface water storage; $F$ (km$^2$) is the area of the basin, and



$\otimes$  is the symbol of convolution.

## 2.4 Model calibration and evaluation

The Shuffled Complex Evolution Algorithm (SCE-UA) technique was developed by the
University of Arizona in 1992 for nonlinear, high dimensional optimization issues (Duan et al., 1992).
The technique has been used extensively for calibrating hydrological models (Zhou et al., 2019).
Consequently, the SCE-UA method was employed in this study to optimize the parameters of the
hydrological model. An aggregated objective function made up of three measures and aimed at
maximizing flow characteristics was used for the parameter calibration (Brunner et al., 2021; Yi et al.,
2022; Yi et al., 2023). The aggregated objective function and three metrics are expressed by
$$E_{\mathrm{NS}} = 1 - \frac{\sum_{t=1}^{T} \left| Q_s^t - Q_o^t \right|}{\sum_{t=1}^{T} \left| Q_o^t - \overline{Q_o} \right|} \tag{16}$$

$$E_{\mathrm{KG}} = 1 - \sqrt{\left(r-1\right)^2 + \left(\frac{\sigma_s}{\sigma_o}-1\right)^2 + \left(\frac{\mu_s}{\mu_o}-1\right)^2} \tag{17}$$

$$R_{\mathrm{SR}} = \sqrt{\frac{\sum_{t=1}^{T} \left(Q_o^t - Q_s^t\right)^2}{\sum_{t=1}^{T} \left(Q_o^t - \overline{Q_o}\right)^2}} \tag{18}$$

$$M = 0.5 \times \left(1 - E_{\mathrm{NS}}\right) + 0.25 \times \left(1 - E_{\mathrm{KG}}\right) + 0.15 \times \left(1 - \log\left(E_{\mathrm{NS}}\right)\right) + 0.1 \times R_{\mathrm{SR}} \tag{19}$$

where  $Q_o^t$  is the observed discharge at time $t$;  $Q_s^t$  is the simulated discharge at time $t$;  $\overline{Q_o}$  is the
mean of the observed discharge; $T$ is the duration of the flood event; $r$ is the correlation coefficient
between the observed and simulated flood; $\sigma_s$ and $\sigma_o$ are the standard deviation values for the simulated
and observed responses, respectively; and $\mu_s$ and $\mu_o$ are the corresponding mean values.
Several criteria were used for the model performance evaluation, consisting of the $E_{\mathrm{NS}}$, the Root
Mean Square Error ($RMSE$), the relative flood peak error ($Q_{\mathrm{P}}$) and the flood peak time error ($T_{\mathrm{P}}$), which
can be expressed by
$$RMSE = \sqrt{\frac{1}{T} \sum_{t=1}^{T} \left(Q_s^t - Q_o^t\right)^2} \tag{20}$$





$$Q_p = \frac{Q_p^s - Q_p^o}{Q_p^o}$$
(21)

$$T_p = T_p^s - T_p^o$$
(22)

where $Q_p^s$ is the simulated flood peak discharge; $Q_p^o$ is the observed flood peak discharge; $T_p^o$ is the
observed flood peak time; and $T_p^s$ is the simulated flood peak time.
## 3. Study area and data

The Longhu River basin and the Dongshi River basin were selected as two case study watersheds.

The Longhu river and the Dongshi river are located in the humid mountain area, which originate from
the Hanjiang River basin of the Guangdong Province, China. The Longhu river is 17.4 km long, with
a basin area of 102.7 km$^2$, and the mean slope of the basin is 2.9 ‰. The Dongshi river is 23.6 km
long, with a basin area of 152.4 km$^2$, and the mean slope of the basin is 3.56 ‰. The DEM data of the
two basins were collected from (http://www.gscloud.cn/). The land cover data can be accessed from
(http://data.ess.tsinghua.edu.cn/). The distributions of the DEM, slope and land cover for the two
basins are shown in Fig. 3.





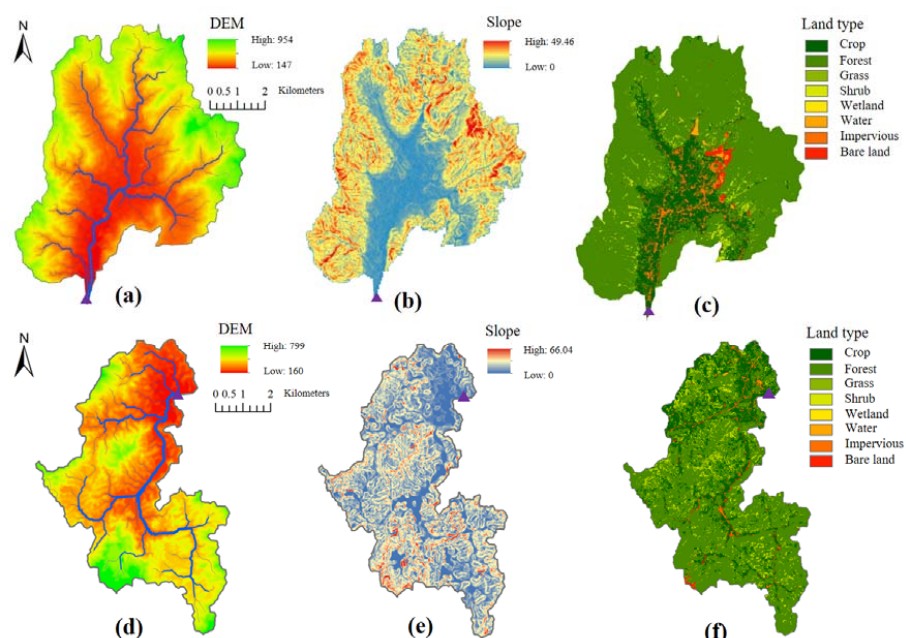

**Figure 3.** Distribution of the DEM, slope and land cover. (a), (b) and (c) are the DEM, slope and land cover corresponding to the Dongshi River basin. (d), (e) and (f) are the DEM, slope and land cover corresponding to the Longhu River basin.

Additionally, the rainfall and evaporation data from meteorological stations for the two basins were collected from 1973 to 2020, and the simultaneous hourly runoff data for the outlet of the watersheds were collected as well. A total of 32 isolated storms with the observed runoff responses from 1973 to 2020 were selected to calibrate and verify the established model. The Dongshi hydrological station was built in recent years, and the flow sequences were therefore short. Specifically, 16 flood events in the Longhu River basin with 1 hour interval were collected from 1973 to 2016, and 16 flood events with 1 hour interval were collected from the Dongshi River basin from 2015 to 2020. The statistics of these flood events are shown in Table 1. The average flood peaks for the Longhu River





basin and the Dongshi River basin are 116.7 m³ s⁻¹ and 73.0 m³ s⁻¹, respectively. The average flood
durations are about 30 h and 33 h, respectively. Moreover, to consider the initial condition, the
antecedent precipitation was calculated based on the daily recession coefficient of the water storage.
**Table 1.** Statistics of the flood events in the Longhu and the Dongshi River basins.

| Watershed | Flood events | Rainfall (mm) | Flood peak (m³ s⁻¹) | Time duration (h) |
|---|---|---|---|---|
| Longhu | 19730508 | 80.0 | 94.5 | 27 |
| | 19730720 | 76.7 | 180.0 | 17 |
| | 19750526 | 54.9 | 101.0 | 21 |
| | 19760702 | 73.0 | 137.0 | 28 |
| | 19770526 | 73.8 | 90.4 | 18 |
| | 19771003 | 62.1 | 97.5 | 19 |
| | 19790607 | 100.3 | 93.4 | 24 |
| | 19890502 | 46.5 | 132.0 | 29 |
| | 20030517 | 94.0 | 140.0 | 46 |
| | 20060601 | 56.0 | 96.5 | 37 |
| | 20120527 | 118.8 | 128.0 | 27 |
| | 20130713 | 214.4 | 228.0 | 30 |
| | 20150601 | 83.4 | 85.0 | 44 |
| | 20150831 | 102.6 | 83.2 | 30 |
| | 20160430 | 111.2 | 91.0 | 54 |
| | 20160903 | 85.4 | 89.7 | 26 |
| Dongshi | 20150509 | 105.2 | 62.9 | 38 |
| | 20150721 | 132.0 | 82.0 | 29 |
| | 20160811 | 90.0 | 51.3 | 48 |
| | 20160819 | 112.5 | 34.9 | 19 |
| | 20161021 | 158.8 | 48.0 | 49 |
| | 20170501 | 84.5 | 98.3 | 22 |
| | 20170515 | 84.0 | 43.7 | 29 |
| | 20170613 | 139.2 | 37.2 | 31 |
| | 20170929 | 71.0 | 101.2 | 25 |
| | 20180606 | 61.5 | 34.9 | 32 |
| | 20180702 | 23.5 | 44.3 | 25 |
| | 20190418 | 86.4 | 35.5 | 18 |
| | 20190609 | 107.6 | 272.0 | 27 |
| | 20190612 | 74.0 | 100.0 | 66 |
| | 20200522 | 67.5 | 71.0 | 37 |
| | 20200607 | 109.3 | 50.6 | 26 |



## 4. Results

### 4.1 Calibration of parameters

#### 4.1.1 Calibrated parameters of the DTDUH flow routing method

The core of the DUH was the calculation of the grid flow velocity. The parameters that need to be determined in Eq. (9) consist of $k$, $S$, $I_c$ and $\gamma$, in which $k$ was the velocity coefficient and was determined based on different underlying surface types or different flow states (Foda et al., 2017), as shown in Fig. 3(c) and Fig. 3(f). $S$ was the grid cell slope of the study areas, which could be obtained from the DEM data of the target basin, as shown in Fig. 3(b) and Fig. 3(e). $I_c$ was determined using hourly mean rainfall intensity of the target basin, and this parameter was $10\,\mathrm{mm\,h^{-1}}$ for the two study watersheds. Additionally, parameter $\gamma$ reflected the influence of soil moisture content on the flow velocity. The parameter $\gamma$ of soil moisture content was determined to be 0.5 to reflect the influence of soil moisture content on the flow velocity for the two basins (Yi et al., 2022). It is noteworthy that the raster size of the two basins were divided into $30\,\mathrm{m} \times 30\,\mathrm{m}$. The rationality of the TDUH in these two basins had been validated in our previous research (Yi and Chen, 2022; Yi et al., 2022), and thus was not calibrated in this study.

#### 4.1.2 Calibrated parameters of runoff generation using the XAJ model

The XAJ model was adopted as the hydrological forecasting framework, in which the surface runoff routing module was replaced by three surface runoff routing methods, and the other modules remained unchanged within the XAJ model. The objective of this study was to explore the influence of spatial heterogeneity of the runoff generation on the UH for flood prediction, and thus the parameters of the runoff generation module kept unchanged. The calibrated and validation processes are as follows:

1) The parameters of the XAJ model were calibrated using the SCEUA method, where the surface



runoff routing method was the linear reservoirs. In total, all the 32 flood events in the Longhu River
basin and the Donshi River basin were used for the calibration of the XAJ + LR (linear reservoir) model
without further dividing the validation period.

2) Then, the TDUH and DTDUH were derived, based on physical characteristics and rainfall

intensities of watersheds. The parameters' determination method is given in Section 4.1.1. In order to
verify the rationality of the TDUH and DTDUH routing methods, we calibrated XAJ+TDUH and
XAJ+DTDUH models separately, and compared their performances with that of XAJ + LR model, as
shown in Appendix (Fig. S1 and Fig. S2). Results show that the proposed method exhibits consistent
or better performances compared with the LR routing method. Specifically, the DTDUH method
performed best over the three methods for the Longhu River basin, and the time to peak error is
significantly better than that of the LR method. While, for the Dongshi River basin, the performances
of the LR method were slightly better than that of the proposed method, which could be the reason that
the DTDUH was computed without the observed runoff, and there are various uncertainties in the
conceptualization or implementation of the DTDUH.

3) Finally, the evapotranspiration module, runoff generation module and runoff separation module

and their parameters obtained in Step 1) were not changed so as to discuss the performance of different
runoff routing models, and the LR routing method was replaced with the TDUH and DTDUH,
respectively. The XAJ model with calibrated parameters in Step 1) and TDUH as well as DTDUH
were used for the validation of the XAJ + (TDUH and DTDUH) model. Since the parameters of the
XAJ model were determined by the model combining with LR method, this calibration method would
be more inclined to optimize the performance of XAJ + LR model. When combined with other runoff





routing models, the accuracy of results may be affected to some extent.

*4.2 Computation of the DTDUH*

The TDUH was derived referring to Yi et al. (2022) and the rationality of the TDUH has been
validated. The derivations of the proposed DTDUH were similar to those of the TDUH. The main
differences lied in that the DTDUH was derived for a specific saturated area. In order to obtain the
DTDUHs corresponding to various saturated states of the watershed, the tension water storage in each
grid cell within the basin was considered to be negatively correlated with the Topographic Wetness
Index (TWI) (Shi et al., 2008; Tong et al., 2018; Yuan et al., 2019). We assumed topographic
information captures the runoff generation heterogeneity at the catchment scale, and the TWI was used
as an index to identify rainfall–runoff similarity (Beven and Kirkby, 1979). Areas with similar TWI
values are regarded as possessing equal runoff generation potential. Specifically, the areas with larger
TWI values tend to be saturated first and contribute to saturation excess rainfall; but the areas with
lower TWI values need more water to reach saturation and generate runoff (Gao et al., 2019).
Then, the proposed distributed unit hydrographs (DTDUH) are computed based on the saturated
areas, which can be expressed by TWI. When calculating discharge using the DTDUH, we should
select various DTDUH based on the time-varying soil moisture within each time interval. Theoretically,
the DTDUH is different at each time interval because $\alpha_t$ is ranging from 0 to 1. However, practically
applying time-varying soil moisture based on DTDUH can be a complex task. In order to improve the
effectiveness of the routing method, the soil moisture contents $\alpha_t$ in Eq. (9) were discretized to 0.25,
0.5, 0.75 and 1 based on the distributions of the TWI. Then, a simplified DTDUH can be obtained in
a certain range soil moisture content. And these ranges are presented in Table 2. The distribution of the





saturated areas corresponding to different soil proportions are shown in Fig. 4. Similarly, the ratio of
excess rainfall intensity and the reference excess rainfall intensity $\frac{I_t}{I_c}$ in Eq. (9) were discretized to
0.5, 1, 1.5 and 2 in order to improve the calculation efficiency. More details can be found in Yi et al.

(2022).

**Table 2.** The soil moisture content $\alpha_t$ of each interval corresponds to the discrete soil moisture $\alpha_s$.

| Soil moisture $\alpha_t$ | $0 \leq \alpha_t \leq 0.25$ | $0.25 < \alpha_t \leq 0.5$ | $0.5 < \alpha_t \leq 0.75$ | $0.75 < \alpha_t \leq 1$ |
|---|---|---|---|---|
| Discrete soil moisture $\alpha_s$ | 0.25 | 0.5 | 0.75 | 1 |


**Figure 4.** Saturated areas corresponding to various soil moisture proportions, where green area
represents the saturated region and white area represents the unsaturated region. (a), (b) and (c) are the
saturated areas corresponding to 0.25, 0.5 and 0.75 soil moisture proportions for the Longhu River
basin. (d), (e) and (f) are the saturated areas corresponding to 0.25, 0.5 and 0.75 soil moisture
proportions for the Dongshi River basin.
The computed TDUHs and DTDUHs for the Longhu River basin and the Dongshi River basin
are demonstrated in Fig. 5 to Fig. 8. For instance, when the excess rainfall intensities range from 0 to
5 mm h$^{-1}$ and the soil moisture contents are between 0 and 0.25, the TDUH used for the surface runoff
routing is corresponding to the red line in Fig. 5(a) for the Longhu River basin. Although the time to
peak and flow peak discharge can be different for various TDUHs, the areas below the curve are the
same with each other. However, the areas below the curve are not necessary the same for the DTDUH,
as the proposed DTDUH is derived from the saturated areas, as shown in Fig. 6. Only when the
watershed reaches a global saturated state will the DTDUH be the same as the TDUH. It is noteworthy



that for the same rainfall intensity, the time to peak of the DTDUHs corresponding to various soil
moisture content are almost consistent with slight variations.

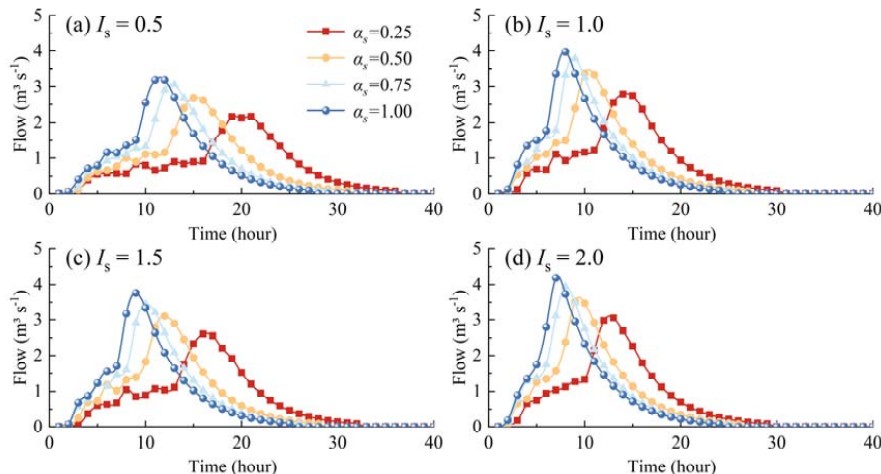


**Figure 5.** The TDUHs of the Longhu River basin. (a) $I_s = 0.5$. (b) $I_s = 1.0$. (c) $I_s = 1.5$. (d) $I_s = 2.0$.

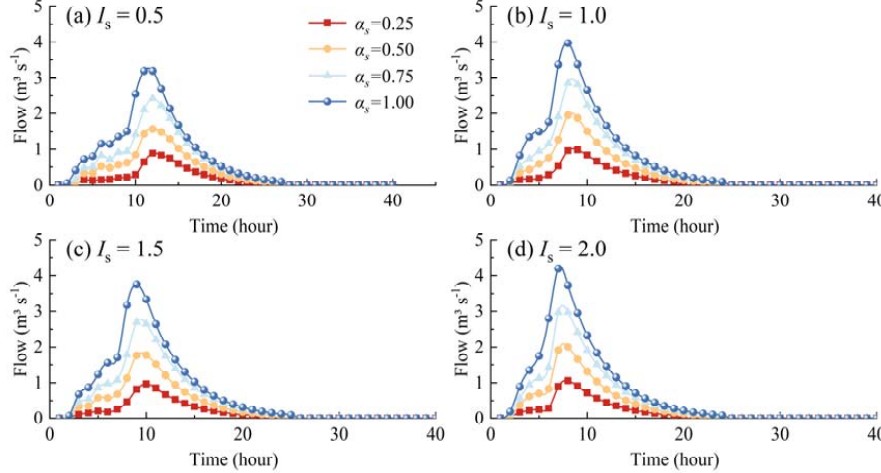


**Figure 6.** The DTDUHs of the Longhu River basin. (a) $I_s = 0.5$. (b) $I_s = 1.0$. (c) $I_s = 1.5$. (d) $I_s = 2.0$.



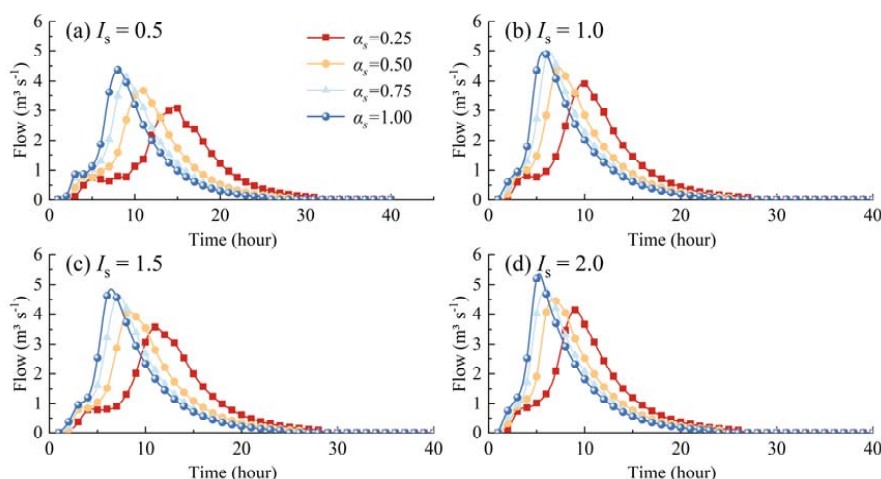

**Figure 7.** The TDUHs of the Dongshi River basin. (a) $I_s = 0.5$. (b) $I_s = 1.0$. (c) $I_s = 1.5$. (d) $I_s = 2.0$.

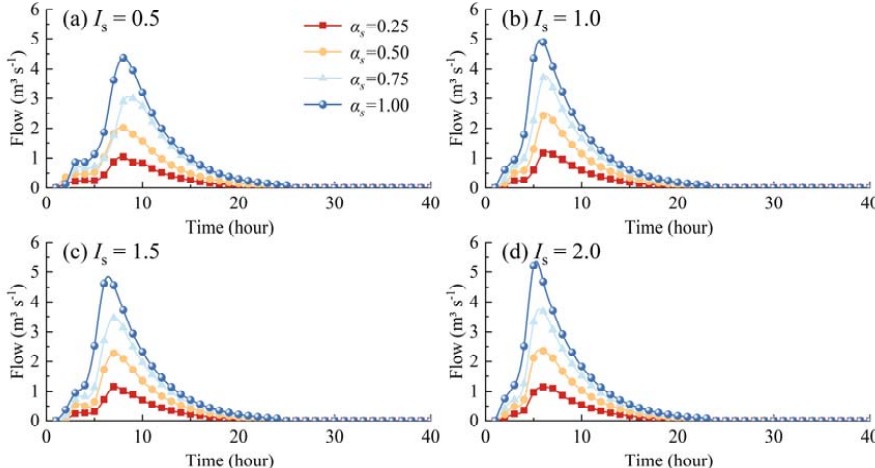

**Figure 8.** The DTDUHs of the Dongshi River basin. (a) $I_s = 0.5$. (b) $I_s = 1.0$. (c) $I_s = 1.5$. (d) $I_s = 2.0$.

*4.3 Errors due to spatial scale mismatch between runoff generation and runoff routing*

To investigate the possible errors due to spatial scale mismatch between runoff generation and runoff routing, we assumed several sets of excess rainfall with intensities of 5, 10, 15 and 20 mm h$^{-1}$, and the saturated proportions of the basin are 0.25, 0.5, 0.75 and 1, respectively. Considering the average rainfall intensities of the two basins, the time durations of the rainfall are assumed changing



from 1 to 4 h, and the combinations are shown in the Table 3. Thus, 16 combinations can be formed
based on Table 3. For example, combination R-T1-S1 indicates there is a total depth of 20 mm excess
rainfall in the global watershed with the time duration being 1 hour (20 mm h$^{-1}$), and the saturated
proportion of the basin is 0.25. Similarly, combination R-T3-S2 indicates there is a total depth of 20
mm excess rainfall in the global watershed with the time durations being 4 hours (5 mm h$^{-1}$), and the
saturated proportion of the basin is 0.50. Then, the flow hydrograph due to the assumed excess rainfall
can be obtained using the TDUH and the DTDUH, thus to compare the errors of spatial scale mismatch
between runoff generation and runoff routing. The flow hydrograph computed using the TDUH and
DTDUH corresponding to the two basins are shown in Fig. 9 and Fig. 10, respectively.
**Table 3.** Combinations of the assumed excess rainfall, time duration and soil moisture content.

| Depth of excess rainfall (mm) | Time duration (h) | Soil moisture content |
|---|---|---|
| 20 (R) | 1 (T1) | 0.25 (S1) |
| 20 (R) | 2 (T2) | 0.50 (S2) |
| 20 (R) | 3 (T3) | 0.75 (S3) |
| 20 (R) | 4 (T4) | 1.00 (S4) |

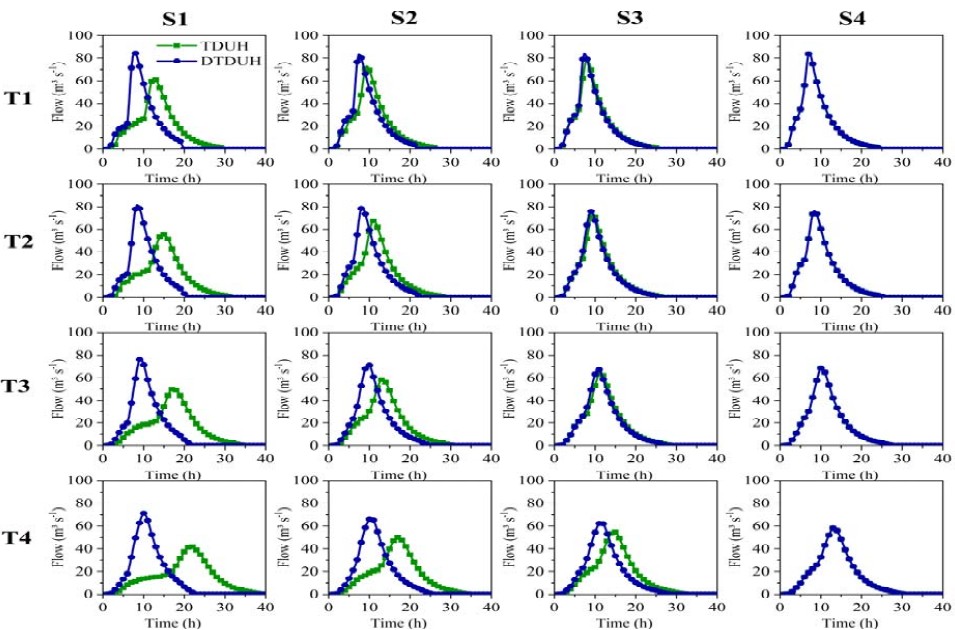


**Figure 9.** Errors of the flow hydrograph due to spatial mismatch between runoff generation and runoff

routing for the Longhu River basin

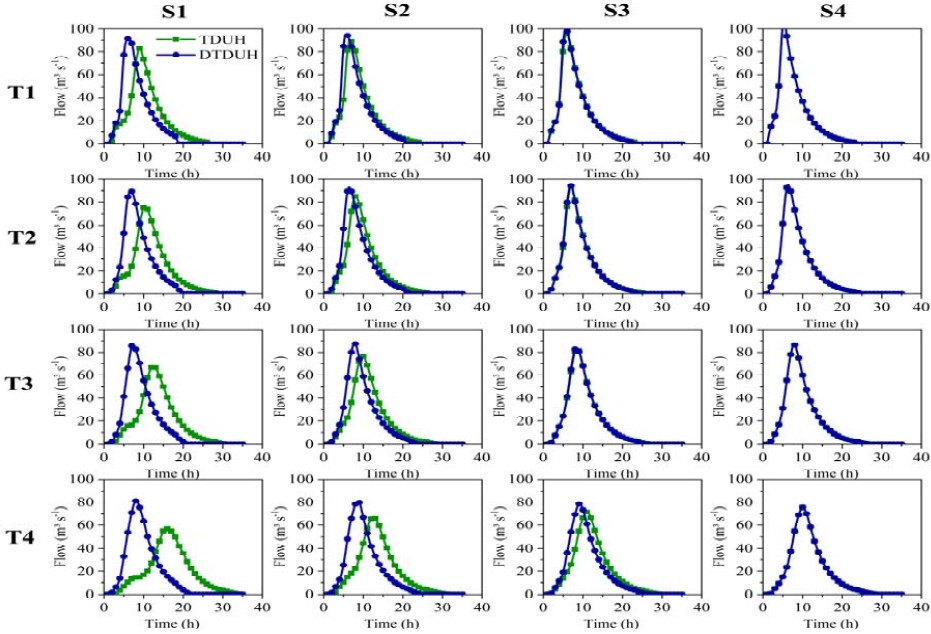




**Figure 10.** Errors of the flow hydrograph due to spatial mismatch between runoff generation and runoff routing for the Dongshi River basin

It can be found from Fig. 9 that the soil moisture significantly influences the results of the TDUH, while the results of the DTDUH are almost not changed with the variation of soil moisture for the Longhu River basin. When the saturated soil moisture proportion is low, the results of the two routing methods are significantly different. Simultaneously, when the saturated soil moisture proportion exceeds 0.5, the results of the TDUH and the DTDUH perform almost consistently. The differences between the results of the two methods increase with the duration of excess rainfall.

For the Dongshi River basin, it can be seen from Fig. 10 that patterns of the soil moisture and time duration on the performances are basically consistent with those of the Longhu River basin. Conversely, the soil moisture shows a more pronounced effect on the DTDUH, and the flow peak discharge become higher with the soil moisture changing from S1 to S4. It can also be found that the differences between the results of the TDUH and the DTDUH in the Dongshi River basin are smaller than those in the Longhu River basin, which means that the two routing methods will perform similarly for the Dongshi River basin.

In summary, the performances of the TDUH and DTDUH are consistent for higher soil moisture and higher rainfall intensity. When the soil moisture is low and the time duration of the excess rainfall is long, we should pay much attention to the errors due to the spatial scale mismatch between runoff generation and runoff routing. Additionally, the differences caused by this mismatch vary significantly in different watersheds.



### 4.4 Performances of the DTDUH for the Longhu River basin


A total of 16 isolated storms with the observed runoff responses from 1973 to 2016 were selected
to explore the performances of the TDUH and DTDUH for the Longhu River basin. Since the TDUH
and the DTDUH were used for the surface runoff routing, the proportions of the surface runoff had
significant influence on the hydrological modelling performances. However, the performances of the
model were evaluated based on the flow hydrograph at the outlet of the watershed, and the flow
hydrograph was composed by three components including the surface runoff, subsurface stormflow
runoff and subsurface runoff. To evaluate the performances of the TDUH and DTDUH under the
condition of XAJ modelling framework, we are supposed to calculate the ratio of surface runoff to the
total depth of excess rainfall, as shown in Fig. 11(a). Results show that the surface runoff accounts for
most of the total depth of excess rainfall, which means that the performances of the hydrological model
are mainly affected by the surface runoff routing methods. To this end, it is reasonable to compare the
performances of the TDUH and DTDUH for the Longhu River basin. Simultaneously, Fig. 11(b) shows
the antecedent soil moisture of the 16 flood events. The proportion of the saturated areas accounts
more than 50% for most of the flood events.

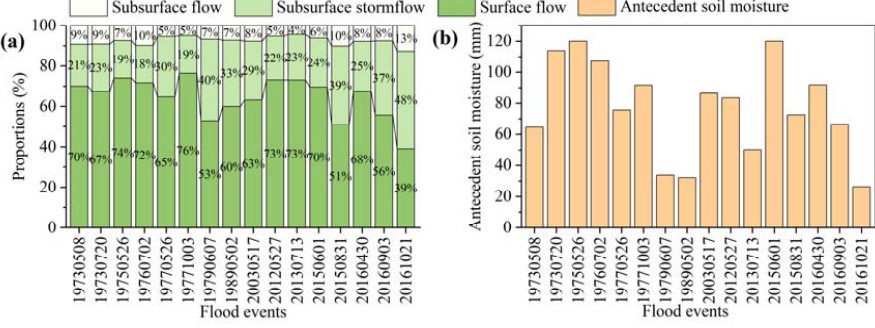


**Figure 11.** Details of the runoff components and antecedent soil moisture of the 16 flood events for
the Longhu River basin. (a) Stacked bar charts of runoff components. (b) Bar charts of the antecedent
soil moisture.

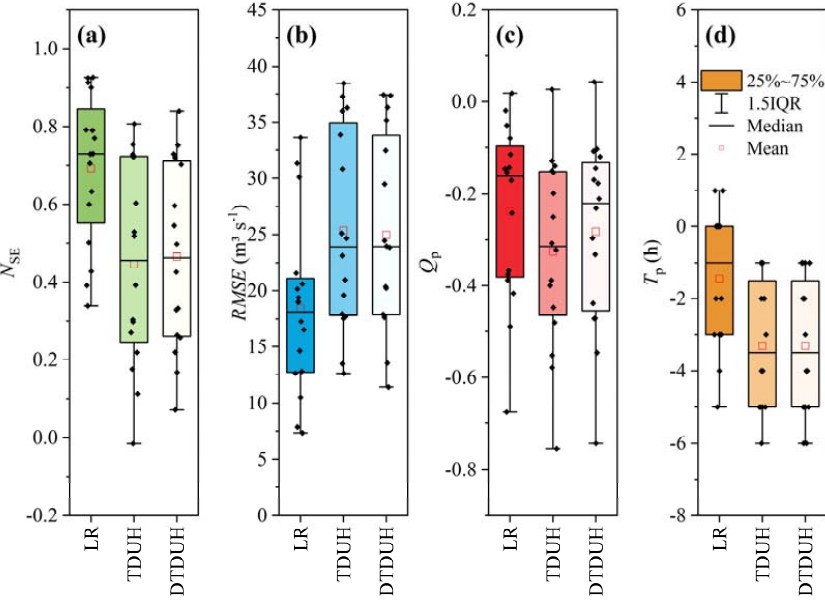


**Figure 12.** Distributions of the evaluation index based on different runoff routing methods for the

Longhu River basin. (a) Distribution of *NSE*. (b) Distribution of *RMSE*. (c) Distribution of $Q_p$. (d)

Distribution of $T_p$.

Three surface runoff routing methods were used for the runoff simulation. Fig. 12 plots the

distributions of the evaluation index using different surface runoff routing methods. Results of the four
indicators show that the average *NSE* of the two DUH methods were low since the parameters of the
XAJ model were calibrated with the linear reservoir method, hence the linear reservoir method
performed the best. The results of the TDUH and DTDUH were basically consistent, and the DTDUH
performed slightly better than TDUH. Noteworthily, it can be seen form Fig. 12(c) that the median of
the relative error of the flood peak of the DTDUH method is increased by 10% compared to that of the





TDUH method, and the average error of the relative flood peak discharge of the proposed method is
significantly smaller than that of the TDUH. The average antecedent soil moisture of the Longhu River
basin is 77 mm which leads to a saturated proportion larger than 0.5. As discussed in the Section 4.2,
the performances of the TDUH and DTDUH are consistent when the saturated proportion is larger
than 0.5.

Additionally, we selected several flood events whose antecedent soil moisture are low and the

*NSE* is larger than 0.5 for analyses. Table 4 lists the evaluation indicators of the three methods in the
runoff forecasting application. It can be seen from Table 4 that all these four flood events show good
performances when the linear reservoir is used for the runoff routing. The antecedent soil moistures of
these flood events are 64, 33, 50 and 26 mm, respectively. The errors of the relative flood peak
discharge of the DTDUH decrease by 3%, 3%, 3% and 8% for the four flood events, compared with
those of the TDUH. The *NSE* of the TDUH and the DTDUH are consistent, and the *RMSE* of the
DTDUH is slightly lower than that of the TDUH methods. The time to flood peak is absolutely the
same for the two methods. It can be concluded that the DTDUH method indicates good performances
for the flood events with low antecedent soil moisture, and it performs consistently with the TDUH
when the global watershed is nearly saturated. This conclusion is consistent with that of the Section
4.2, and the assumption that there will be considerable errors due to the spatial scale mismatch between
runoff generation and runoff routing in low antecedent soil moisture, has been validated.
**Table 4.** Indicators of different routing methods for the Longhu River basin.

| Indicators | Flood events | Linear reservoir | TDUH | DTDUH | Performances |
|---|---|---|---|---|---|
| *NSE* | 19730508 | 0.93 | 0.81 | 0.84 | ↑ |
|  | 19790607 | 0.73 | 0.60 | 0.60 | − |





| | | | | | |
|---|---|---|---|---|---|
| | 20130713 | 0.91 | 0.72 | 0.70 | ↓ |
| | 20161021 | 0.90 | 0.72 | 0.72 | − |
| | 19730508 | 7.85 | 12.62 | 11.46 | ↑ |
| *RMSE* | 19790607 | 14.63 | 17.71 | 17.85 | ↓ |
| | 20130713 | 20.15 | 36.03 | 37.36 | ↓ |
| | 20161021 | 10.51 | 17.55 | 17.64 | ↓ |
| | 19730508 | -0.15 | -0.20 | -0.17 | ↑ |
| $Q_p$ | 19790607 | -0.14 | -0.14 | -0.11 | ↑ |
| | 20130713 | -0.05 | -0.15 | -0.12 | ↑ |
| | 20161021 | -0.08 | -0.25 | -0.17 | ↑ |
| | 19730508 | 0 | -4 | -4 | − |
| $T_p$ | 19790607 | 0 | -2 | -2 | − |
| | 20130713 | 0 | -1 | -1 | − |
| | 20161021 | -3 | -5 | -5 | − |

*Note: The symbols ↑, ↓ and − mean the performance of the DTDUH is better, lower and*
*consistent compared with that of the TDUH.*
*4.5 Performances of the DTDUH for the Dongshi River basin*

Similarly, a total of 16 isolated storms with the observed runoff responses from 2015 to 2020

were selected to explore the performances of the TDUH and DTDUH for the Dongshi River basin.
The components of the runoff and the antecedent soil moisture for the 16 flood events are shown in
Fig. 13(a) and Fig. 13(b). Fig. 13(a) shows that the depth of the surface runoff accounts for more than
70% of the total depth runoff and the study watershed has no subsurface stormflow, which means that
the Dongshi River basin can also be well used for exploring the performances of the TDUH and the
DTDUH. The average antecedent soil moisture of the 16 flood events is 87 mm, which is larger than
that of the Longhu River basin.

Fig. 14 plots the distributions of the evaluation index using different flow routing methods for the

Dongshi River basin. It can be seen from Fig. 14 that the linear reservoir method shows the best
performances overall, and the TDUH and DTDUH have consistent performances. The consistent
performances of the TDUH and DTDUH can be due to the fact that the average antecedent soil
moisture of these 16 flood events is high. Hence, the errors caused by the spatial scale mismatch are
small.

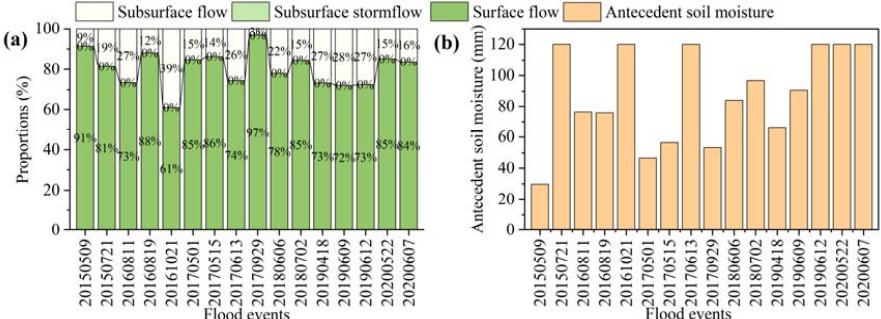


**Figure 13.** Details of the runoff components and antecedent soil moisture of the 16 flood events for
the Dongshi River basin. (a) Stacked bar charts of runoff components. (b) Bar charts of the antecedent
soil moisture.

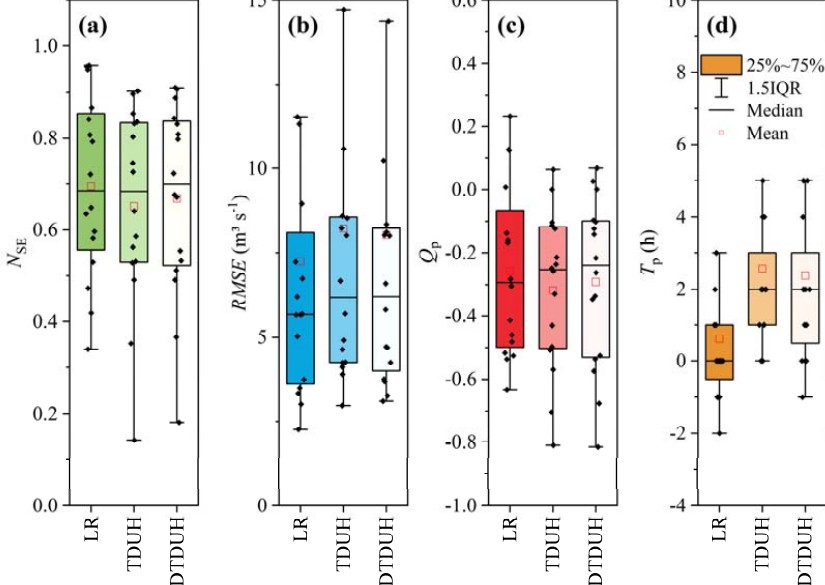


**Figure 14.** Distributions of the evaluation index based on different runoff routing methods for the





Dongshi River basin. (a) Distribution of *NSE*. (b) Distribution of *RMSE*. (c) Distribution of $Q_p$. (d)
Distribution of $T_p$.

However, the differences of the results of the TDUH and DTDUH can be identified in specific

flood event. Table 5 lists the evaluation indicators of the three methods in the runoff forecasting
application for flood events No. 20170501, No. 20170515, No. 20170929 and No. 20190418. The
antecedent soil moistures of these flood events are 47, 56, 53 and 66 mm, respectively. It can be found
from Table 5 that the performances of the DTDUH are better than those of the TDUH. Therefore, we
can conclude that the proposed DTDUH shows higher accuracy for the flood events with lower
antecedent soil moisture, and these results are the same with those of the Longhu River basin.
**Table 5.** Indicators of different routing methods for the Dongshi River basin.

| Indicators | Flood events | Linear reservoir | TDUH | DTDUH | Performances |
|---|---|---|---|---|---|
| *NSE* | 20170501 | 0.87 | 0.64 | 0.67 | ↑ |
| | 20170515 | 0.72 | 0.59 | 0.68 | ↑ |
| | 20170929 | 0.42 | 0.73 | 0.81 | ↑ |
| | 20190418 | 0.64 | 0.53 | 0.55 | ↑ |
| *RMSE* | 20170501 | 3.01 | 4.91 | 4.71 | ↑ |
| | 20170515 | 3.49 | 4.25 | 3.75 | ↑ |
| | 20170929 | 5.67 | 3.90 | 3.25 | ↑ |
| | 20190418 | 7.24 | 8.23 | 8.01 | ↑ |
| $Q_p$ | 20170501 | -0.28 | -0.26 | -0.10 | ↑ |
| | 20170515 | 0.12 | -0.12 | -0.10 | ↑ |
| | 20170929 | 0.23 | -0.43 | 0.067 | ↑ |
| | 20190418 | -0.52 | -0.70 | -0.68 | ↑ |
| $T_p$ | 20170501 | 0 | 4 | 0 | ↑ |
| | 20170515 | 0 | 4 | 4 | − |
| | 20170929 | -2 | 2 | 1 | ↑ |
| | 20190418 | 1 | 2 | 2 | − |

*Note: The symbols  ↑ ,  ↓  and  −  mean the performance of the DTDUH is better, lower and*
*consistent compared with that of the TDUH.*





## 5. Discussion

### 5.1 Forecasting performance advantage analysis of the proposed DTDUH

There is no issue on the spatial scale mismatch between runoff generation and runoff routing in the fully distributed hydrological model, since the fully distributed model solves the St. Venant equations, however, they are usually computationally intensive (Bunster et al., 2019). For the conceptual models, the spatially DUH method is an alternative method that allows the use of distributed information in a much more efficient manner. A novel DUH method was proposed in this study for the surface runoff routing, namely the DTDUH. The proposed DTDUH was computed based on the runoff generation areas instead of the whole basin. This realization was significant different with the current understandings (Gibbs et al., 2010; Goñi et al., 2019; Andrieu et al., 2021). To this end, the TDUH and the DTDUH can be used for exploring the influence of spatial heterogeneity of runoff generation on the runoff routing. The differences of the definition, assumptions as well as limitations between the current theory and the DTDUH have been concluded in Table 6.

**Table 6.** The differences of the definition, assumptions as well as limitations between the current theory and the DTDUH.

| Differences | Current theory | DTDUH |
|---|---|---|
| Definition | A typical hydrograph of direct runoff which gets generated from one centimetre of effective rainfall falling at a uniform rate over the entire drainage basin uniformly during a specific duration. | A typical hydrograph of direct runoff which gets generated from one centimetre of effective rainfall falling at a uniform rate over the saturated drainage basin uniformly during a specific duration. |
| Assumption | i) The effective rainfall is uniformly distributed over the entire drainage basin. ii) The effective rainfall occurs uniformly within its specifier duration. iii) The principle time invariance. | i) The effective rainfall is uniformly distributed over the saturated drainage areas. ii) The effective rainfall occurs uniformly within its specifier duration. |



| | | iii) It can be time variant. |
|---|---|---|
| Limitation | i) The limiting size of the drainage basin is considered to be 5000 km$^2$. ii) The physical basin characteristics must be unchanged. | i) The limiting size of the drainage basin is small watershed, otherwise the DUH method is not applicable. ii) The saturated areas should be obtained based on the physical characteristics of the watershed. |

In this study, the DTDUH was computed considering the time-varying characteristics of the
saturated areas, and the excess rainfall was redistributed in the saturated areas instead of the whole
basin. The diagram of the DTDUH derivation processes corresponding to various saturated soil
moisture can be shown as Fig. 15. The total travel time from each saturated grid cell to the outlet is
obtained by directly recording each particle's total travel time from the initial location until the particle
leaves the basin. We can obtain the DTDUH when the last particle leaves the basin. There are 24 grid
cells in the basin. For instance, the derivation of the TDUH corresponding to saturated proportions 25%
(6 grid cells) is shown as Fig. 15(a). The DTDUH is not the same with the TDUH until the basin reach
a global saturation as Fig. 15(d).

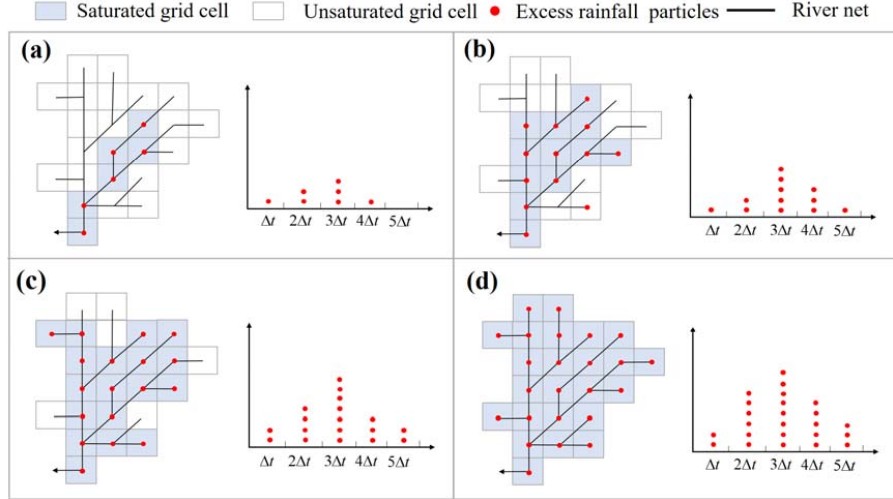


**Figure 15.** The diagram of the DTDUH corresponding to various soil moisture. (a) $\alpha_s = 0.25$. (b)





$\alpha_s = 0.50$. (c) $\alpha_s = 0.75$. (d) $\alpha_s = 1.00$.
For instance, when there occurs a stormflow with the depth of 10 mm in the whole basin and the
saturated proportion is 50%, the actual depth of the excess rainfall in the saturated area is 20 mm. In
tradition, the flow hydrograph was calculated using the TDUH as Fig 15. (d), neglecting the issues that
excess rainfall particles are only on the saturated areas. This neglection therefore leads to errors
because the unsaturated areas where no excess rainfall is generated contributes to the confluence. To
solve this issue, the flow hydrograph was calculated using the DTDUH as Fig. 15 (b) in this study, and
in turn can improve the forecast performances of the hydrological model.

### 5.2 Potential shortcomings and improvements of the proposed DTDUH

Although the performances of the DTDUH methods are better than those of the TDUH methods,
there are still some issues that need to be addressed: 1) A hybrid runoff generation process pattern
formed by more than one mechanism can often be identified in semi-humid, semi-arid and mountain
watershed, because of the heterogeneity of underlying surface conditions and meteorological factors
(Hu et al., 2021; Yi et al., 2023). When there occurs more than the saturation-excess rainfall, the
saturated area extraction method based on the TWI will not be applicable as the excess rainfall can
also be generated from the unsaturated areas; 2) As discussed above, the saturation areas were extracted
based on the TWI in this study (Tong et al., 2018; Yuan et al., 2019). When the rainfall is spatially
uneven distributed over the watershed, the errors of the extracted saturated area can be huge as the gird
cells with larger tension water capacity may be saturated in a shorter time than those with smaller
tension water capacity. Thus, determining the time-varying saturated grid cells is a considerable
challenge for the accurate hydrological forecasting, since it influences the accuracy of the DTDUH; 3)





The DTDUH can be well used for the areas with abundant surface runoff. For the semi-humid and
semi-arid watershed with a large proportion of the subsurface stormflow runoff and subsurface runoff,
how to develop a dynamic confluence method suitable for these regions is another challenging
hydrologic issue because the mechanism of the subsurface stormflow generation is ambiguous, and
quantifying the subsurface stormflow generating processes is a formidable task as the subsurface
stormflow runoff and subsurface runoff cannot be observed directly.
**6. Conclusions**
A novel DTDUH method was proposed to explore the influence of spatial heterogeneity of runoff
generation on the runoff routing. The XAJ model was used as the hydrological modelling framework.
The Longhu River basin and the Dongshi River basin were selected as two case studies. The results of
the three surface runoff routing methods including the linear reservoir, TDUH and DTDUH were
compared. The advantages and shortcomings of the proposed method have been discussed. The main
conclusions can be summarized as follows:
1. A novel DTDUH method designed for the surface runoff routing was proposed based on the
TDUH method. The traditional TDUH method was derived based on the whole basin, and the proposed
method was designed only for the saturated areas. The DTDUH method considered not only the time-
varying rainfall intensities and soil moisture, but also the time-varying saturated areas of the watershed
which were extracted based on the TWI.
2. The rationality of the proposed method was verified by comparing the performances of
XAJ+LR, XAJ + TDUH and XAJ +DTDUH models, which were calibrated separately. Results shows
that the proposed method exhibited consistent or better performance compared with that of the LR





routing method, and performed better than the TDUH method.

3. The influence of spatial heterogeneity of runoff generation on the runoff routing was carried

out by comparing the performances of the TDUH and the DTDUH. The performances of the proposed
DTDUH has been compared with the traditional TDUH method in the Longhu River basin and the
Dongshi River basin while the parameters of the runoff generation module remained the same. Results
show that the DTDUH show a better performance for the flood events with low antecedent soil
moisture, and when the antecedent soil moisture is high, the performance of the two methods
performed consistently.

4. The differences of definition, assumptions as well as limitations between the current UH theory

and the DTDUH have been discussed. Additionally, some challenges related to the DTDUH has been
presented, for example, the hybrid runoff being generated more than in the saturated areas, the
extraction accuracy of the time-varying saturation areas, and the spatial scale mismatch between runoff
generation and runoff routing of the subsurface stormflow runoff as well as the subsurface runoff.



**Data availability**

Due to the strict security requirements from the departments, some or all data, models,

or code generated or used in the study are proprietary or confidential in nature and may

only be provided with restrictions (e.g. anonymized data).

**Author contributions**

Lu Chen conceived the original idea, and Bin Yi designed the methodology, developed

the code and performed the study. Lu Chen, Bin Yi and Tao Xie contributed to the

interpretation of the results. Bin Yi wrote the paper, and Lu Chen revised the paper.

**Competing interests**

The authors declare that they have no conflict of interest.

**Acknowledgments**

This study was financially supported by the National Key R&D Program of China

(2023YFC3081000, 2021YFC3200400), and the Science and Technology Plan Projects

of Tibet Autonomous Region (XZ202301YD0044C).

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
