# Peer review of "On the Influence of Spatial Heterogeneity of Runoff Generation on the"

_Hydrology and Earth System Sciences, 2024_

## Author Comment (AC1)

Dear Reviewer #2,

The authors would like to thank the reviewer for your time and constructive comments. Our point-to-point responses are listed below, where our responses are in blue, and the reviewers' comments are in black.

Kind regards, all authors

**General comments**

This paper deals with rainfall-runoff modelling and proposes a new routing scheme adapted from the principle of geomorphological and distributed unit hydrographs (Rodriguez-Iturbe and Valdès, 1979; Maidment et al., 1996). The new proposed scheme is called Dynamic Time Varying Unit Hydrographs, since it combines i) a variation of flow velocities depending on the rainfall intensities and the saturation level of the basin, and ii) a variation of the flow contributing areas considered to generate the unit hydrograph, which are assumed to correspond to the saturated areas. This basic assumption of runoff dominated by saturated areas of course limits the potential of application of the method (contributions from saturated areas should dominate the other flow generation processes), but this limitation is clearly stated in the conclusions of the manuscript, and this assumption seems to be valid in the presented case studies (humid basins).

The topic of this paper is very relevant to HESS, and the proposed method brings significant novelty. Unfortunately, I think that the presentation of the methods and results should still be largely improved. Moreover, the methodology proposed for evaluation does not appear relevant to me, and does not sufficiently support the conclusions of this work, in my opinion. Particularly, the manuscript should be improved on the following aspects:

**Comment 1:**The description of the hydrological model is not sufficiently clear (consistency and links between equations, variables and parameters, description of parameters to be calibrated). See my detailed remarks below

**Response:**

We are sorry for the unclear description of the methodology. We will rewrite this part, and two pictures of the tension water storage capacity and free water storage capacity curves will be added to clarify the modelling process of the XAJ model. The revised version is as follows:

[Figure]

**Figure 1.** (a) Tension water storage capacity curve, and (b) Free water storage capacity curve

In the XAJ model, two parabolic curves are adopted to represent the spatially non-uniform distribution of the tension water storage and free water storage. The difference between field capacity and total soil water content is defined as tension water storage capacity, the maximum amount of water available in unsaturated zone. In the XAJ model, a tension water capacity curve is introduced (Fig. 1(a)) to describe the non-uniform distribution of tension water capacity throughout basin or sub-basin. In Fig. 1(a), $A_{ps}$ is the partial pervious area where the tension water storage capacity is less than or equal to the value $W$, which is the tension water capacity at a point, varying from 0 to a maximum $WMM$; $A_p$ is the pervious area; $W_0$ is the initial areal mean tension water storage (mm); and $AU$ is the vertical coordinate corresponding to $W_0$. The functional relationship of the tension water storage capacity curve is expressed as:

$$\frac{A_{ps}}{A_p} = 1 - \left(1 - \frac{W}{WMM}\right)^B \tag{1}$$

Based on Eq. (1), when rainfall exceeds evaporation, the runoff generated in the saturated areas can be calculated as:

$$R = \begin{cases} \int_{AU}^{AU+PE} \frac{A_{ps}}{A_p} dW & PE + AU < WMM \\ PE - WM + W_0 & PE + AU \geq WMM \end{cases} \tag{2}$$

The total runoff $R$ ($R=RS+RI+RG$), generated in a wet period in accordance with Eq. (2), must be separated into three components, including the surface runoff, the subsurface stormflow and the surface runoff. Thus, the concept of free water storage capacity was used, and it was assumed to be distributed between zero and a point maximum $SMM$ in a parabolic manner, as shown in Fig. 1(b). In Fig. 1(b), $A_p$ is the pervious area of the catchment; $A_f$ is the area where the free water storage capacity is less than or equal to the value $S$, varying from 0 to $SMM$; $A_s$ is the runoff generation area; $S_0$ is the initial areal mean free water storage (mm); $BU$ is the vertical coordinate corresponding to $S_0$; and $RI$, $RG$ represent the depth of the interflow and subsurface flow. The functional relationship of the free water storage capacity curve is expressed as:

$$\frac{A_f}{A_s} = 1 - \left(1 - \frac{S}{SMM}\right)^{EX} \tag{3}$$

The total runoff $R$, generated in a wet period in accordance with Eq. (2), can be subsequently separated into three components, including the surface runoff, interflow and groundwater, which can be given by

$$RS = \begin{cases} FR\int_{BU}^{BU+PE} \frac{A_f}{A_s} dS & PE + BU < SMM \\ FR(PE + S_0 - SM) & PE + BU \geq SMM \end{cases} \tag{4}$$

$$RI = KI \cdot FR \cdot \int_0^{BU} \frac{A_f}{A_s} \mathrm{d}S \qquad (5)$$

$$RG = KG \cdot FR \cdot \int_0^{BU} \frac{A_f}{A_s} \mathrm{d}S \qquad (6)$$

where $RS$, $RI$, $RG$ represent the depth of the surface runoff, interflow and groundwater respectively (mm); $FR$, equaling to $R/PE$, is the proportion of the runoff producing area over the whole basin; $SM$ is the areal mean free water capacity (mm); and $KI$ and $KG$ are outflow coefficients of the free water storage to interflow and groundwater, respectively.

**Comment 2:** The principle to replace only the routing scheme for the surface runoff part of the model is presented too late. This should be stated at the beginning, and the routing schemes used for other (subsurface) components should be at least quickly presented.

**Response:**

Thank you for your comments. We will move this statement at the beginning of the manuscript, and the routing schemes for the subsurface stormflow and subsurface runoff will be added in the revised manuscript. The revised sentences are as follows:

For the runoff routing module in the XAJ model, the subsurface stormflow ($RI$) and subsurface runoff ($RG$) were considered using a free reservoir, but the proposed routing schemes are applied only on RS. Their expressions are given by

$$QI_t = CI \cdot QI_{t-1} + (1 - CI) \cdot RI_t \cdot U \qquad (7)$$

$$QG_t = CG \cdot QG_{t-1} + (1 - CG) \cdot RG_t \cdot U \qquad (8)$$

**Comment 3:** The methodological choice to use only the XAJ+LR calibration for the evaluation of the XAJ+TDUH and XAJ+DTDUH models does not appear relevant to me. This leads logically to better performances of the XAJ+LR models, and may also advantage one of the two other XAJ+TDUH and XAJ+DTDUH models. The calibration results of the XAJ+TDUH and XAJ+DTDUH models (appendix) suggest that DTDUH does not perform as well as expected. I think that to achieve robust conclusions all the models should be calibrated and evaluated in validation (with a comparison of calibrated parameters).

**Response:**

Thank you for your comments. We will calibrate the three models (XAJ+LR, XAJ+TDUH and XAJ+DTDUH), respectively, and these results will be added in the revised manuscript.

Table 3 lists the $NSE$, $RMSE$, error of flood peak ($Q_p$) and error of time to peak ($T_p$) values for all the models of the Longhu River basin, among which No. 20150831, 20160430, 20160903 and 201651021 are validation dataset. To demonstrate the model performances of different strategies more visually, Fig. 4 shows line charts of the three runoff routing methods for the four indexes in both calibration and validation periods. Results shows that TDUH and

DTDUH have consistent performances in *NSE*, *RMSE* and $T_p$, and results of the LR method are not stable, sometimes better than that of the DTDUH method, sometimes worse (especially for the criteria $Q_p$). In the calibration periods, DTDUH performs better than that of the TDUH, while in the validation periods, TDUH and DTDUH perform almost consistent in *RMSE*, $Q_p$ and $T_p$. It may be due to the initial conditions of flood events are different from each other. For the calibration periods, the average *NSE* of the LR, TDUH, DTDUH methods is 0.58, 0.61 and 0.64, respectively, and 0.80, 0.78 and 0.82 for the validation periods. Simultaneously, the absolute error of the flood peak is 0.36, 0.25 and 0.24 for the calibration periods, respectively, and 0.18, 0.06 and 0.03 for the validation periods. In general, the improvement from the DTDUH is small, and we summarized the possible reasons as follows 1) The surface runoff accounts for 60% ~ 70% of the total runoff, and it is necessary to consider the influence of the heterogeneity of the subsurface stormflow as well as the subsurface runoff; 2) the antecedent soil moisture is high, and, thus, the simulation error caused by the spatial heterogeneity of runoff generation is small.

**Table 3**. Calibrated and Validated results of the Longhu River basin.

| Flood events | *NSE* | | | *RMSE* (m³/s) | | | $Q_p$ | | | $T_p$ (h) | | |
|---|---|---|---|---|---|---|---|---|---|---|---|---|
| | LR | TDUH | DTDUH | LR | TDUH | DTDUH | LR | TDUH | DTDUH | LR | TDUH | DTDUH |
| 19730508 | 0.77 | 0.83 | 0.85 | 13.66 | 11.67 | 11.11 | -0.28 | -0.22 | -0.21 | 3 | 3 | 3 |
| 19730720 | 0.84 | 0.87 | 0.90 | 22.09 | 19.85 | 17.69 | -0.22 | -0.05 | -0.05 | 1 | 0 | 0 |
| 19750526 | 0.60 | 0.76 | 0.86 | 17.04 | 13.24 | 10.13 | -0.36 | -0.27 | 0.18 | 0 | -1 | 0 |
| 19760702 | 0.59 | 0.68 | 0.67 | 21.96 | 19.45 | 19.73 | -0.53 | -0.39 | -0.39 | 1 | 3 | 3 |
| 19770526 | 0.38 | 0.11 | 0.10 | 21.83 | 26.09 | 26.28 | -0.30 | -0.19 | -0.19 | 3 | 2 | 2 |
| 19771003 | 0.54 | 0.32 | 0.37 | 19.29 | 23.47 | 22.60 | -0.28 | -0.20 | -0.19 | 0 | 3 | 2 |
| 19790607 | 0.31 | 0.34 | 0.47 | 23.41 | 22.91 | 20.39 | -0.28 | -0.25 | -0.25 | 2 | 2 | 2 |
| 19890502 | 0.46 | 0.49 | 0.52 | 28.24 | 27.65 | 26.80 | -0.64 | -0.66 | -0.50 | -1 | 1 | 0 |
| 20030517 | 0.49 | 0.54 | 0.54 | 31.63 | 30.30 | 30.07 | -0.35 | -0.32 | -0.32 | 0 | -1 | -1 |
| 20120527 | 0.72 | 0.73 | 0.72 | 21.15 | 20.69 | 21.03 | -0.25 | 0.03 | 0.03 | -2 | 0 | 0 |
| 20130713 | 0.70 | 0.82 | 0.81 | 37.50 | 28.74 | 30.22 | -0.33 | -0.30 | -0.32 | 1 | 0 | 0 |
| 20150601 | 0.52 | 0.84 | 0.82 | 19.08 | 10.94 | 11.70 | 0.50 | 0.15 | 0.25 | 0 | -1 | -1 |
| 20150831 | 0.82 | 0.74 | 0.76 | 11.88 | 14.12 | 13.58 | -0.21 | -0.02 | 0.00 | -3 | 1 | 1 |
| 20160430 | 0.93 | 0.72 | 0.77 | 7.33 | 13.29 | 13.12 | 0.01 | 0.11 | 0.08 | 1 | 1 | 1 |
| 20160903 | 0.70 | 0.76 | 0.81 | 16.67 | 13.20 | 13.75 | -0.36 | -0.09 | -0.04 | -4 | 0 | 0 |
| 20161021 | 0.73 | 0.90 | 0.94 | 17.22 | 8.47 | 8.50 | -0.16 | 0.03 | -0.01 | -2 | -1 | -1 |

[Figure]

[Figure]

[Figure]

**Figure 4.** Line chart of the *NSE*, *RMSE*, $Q_p$ and $T_p$ for the Longhu River basin.

Simultaneously, Table 4 lists the *NSE*, *RMSE*, error of flood peak ($Q_p$) and error of time to peak ($T_p$) values for all the models of the Dongshi River basin, among which No. 20190609, 20190612, 20200522 and 20200607 are validation dataset. To demonstrate the model performances of different strategies more visually, Fig. 5 shows line charts of the three runoff routing methods for the four indexes in both calibration and validation periods. Compared with the results of Longhu River, the DTDUH and LR methods shows consistent performances, and significantly better than that of the TDUH method. It shows that the DTDUH method exhibits significant improvement in this basin.

**Table 4**. Calibrated and validated results of the Dongshi River basin.

| Flood events | NSE | | | RMSE (m³/s) | | | $Q_p$ | | | $T_p$ (h) | | |
|---|---|---|---|---|---|---|---|---|---|---|---|---|
| | LR | TDUH | DTDUH | LR | TDUH | DTDUH | LR | TDUH | DTDUH | LR | TDUH | DTDUH |
| 20150509 | 0.37 | 0.09 | 0.32 | 12.61 | 15.17 | 13.08 | -0.73 | -0.83 | -0.76 | 5 | 5 | 5 |
| 20150721 | 0.81 | 0.87 | 0.80 | 11.21 | 9.25 | 11.60 | 0.02 | -0.16 | 0.06 | -1 | -1 | -1 |
| 20160811 | 0.61 | 0.54 | 0.67 | 5.17 | 5.63 | 4.78 | -0.34 | -0.17 | -0.38 | 1 | 1 | 1 |
| 20160819 | 0.83 | 0.68 | 0.79 | 2.39 | 3.34 | 2.71 | -0.26 | -0.40 | -0.26 | 0 | 0 | 0 |
| 20161021 | 0.93 | 0.81 | 0.94 | 6.99 | 11.80 | 6.75 | -0.11 | -0.27 | -0.10 | 2 | 5 | 2 |
| 20170501 | 0.89 | 0.67 | 0.86 | 2.69 | 4.75 | 3.11 | -0.16 | -0.27 | 0.00 | 0 | 1 | 0 |
| 20170515 | 0.64 | 0.78 | 0.64 | 3.95 | 3.10 | 3.97 | -0.05 | -0.14 | -0.07 | 0 | 4 | 4 |
| 20170613 | 0.91 | 0.76 | 0.91 | 8.40 | 13.48 | 8.28 | -0.26 | -0.38 | -0.22 | 2 | 2 | 2 |
| 20170929 | 0.75 | 0.53 | 0.73 | 3.71 | 5.12 | 3.87 | -0.20 | -0.21 | 0.17 | -3 | 2 | 0 |
| 20180606 | 0.55 | 0.34 | 0.53 | 5.53 | 6.73 | 5.68 | -0.51 | -0.55 | -0.55 | 1 | 1 | 1 |
| 20180702 | 0.55 | 0.59 | 0.48 | 4.17 | 3.96 | 4.48 | -0.51 | -0.45 | -0.58 | 2 | 2 | 2 |
| 20190418 | 0.61 | 0.55 | 0.58 | 7.54 | 8.00 | 7.79 | -0.63 | -0.67 | -0.65 | 2 | 12 | 2 |
| 20190609 | 0.55 | 0.56 | 0.75 | 13.27 | 13.03 | 9.83 | -0.54 | -0.53 | -0.42 | 2 | 2 | 2 |
| 20190612 | 0.81 | 0.76 | 0.83 | 4.57 | 5.09 | 4.36 | -0.18 | -0.06 | -0.02 | 0 | 0 | 0 |
| 20200522 | 0.77 | 0.56 | 0.75 | 5.24 | 7.32 | 5.56 | -0.01 | -0.02 | -0.04 | 2 | 2 | 2 |
| 20200607 | 0.57 | 0.54 | 0.61 | 28.85 | 30.03 | 27.54 | -0.50 | -0.54 | -0.50 | 0 | -1 | 2 |

[Figure]

**Figure 5.** Line chart of the *NSE*, *RMSE*, $Q_p$ and $T_p$ for the Dongshi River basin.

In addition, the calibrated parameter sets for the two basins are given in Tables 5, and comparisons will also be added in the revised version.

**Table 5**. Calibrated parameters of the three runoff routing methods for the Longhu and Dongshi River basins

| Parameters | Longhu | | | Dongshi | | |
|---|---|---|---|---|---|---|
| | LR | TDUH | DTDUH | LR | TDUH | DTDUH |
| *UM* | 9.65 | 7.13 | 8.29 | 5.38 | 8.16 | 9.13 |
| *LM* | 86.32 | 85.97 | 81.23 | 85.94 | 66.54 | 85.21 |
| *DM* | 43.96 | 47.26 | 49.25 | 47.14 | 28.53 | 45.65 |
| *B* | 0.13 | 0.39 | 0.36 | 0.40 | 0.40 | 0.40 |
| *IM* | 0.48 | 0.49 | 0.49 | 0.26 | 0.02 | 0.20 |
| *KC* | 0.12 | 0.49 | 0.80 | 1.48 | 1.50 | 1.44 |
| *C* | 0.12 | 0.12 | 0.12 | 0.16 | 0.15 | 0.12 |
| *SM* | 23.93 | 33.84 | 35.98 | 50.00 | 50.00 | 50.00 |
| *EX* | 1.19 | 1.24 | 1.10 | 1.00 | 1.00 | 1.00 |
| *KI* | 0.63 | 0.43 | 0.41 | 0.17 | 0.11 | 0.13 |
| *KG* | 0.07 | 0.27 | 0.29 | 0.53 | 0.59 | 0.57 |
| *CI* | 0.20 | 0.51 | 0.56 | 0.51 | 0.52 | 0.49 |
| *CG* | 0.94 | 0.95 | 0.94 | 0.99 | 0.99 | 0.99 |
| *CS* | 0.99 | - | - | 1.00 | - | - |
| *L* | 0.00 | 0.00 | 0.00 | 0.00 | 0.00 | 0.00 |

**Comment 4:**Some of the presented results appear to be inconsistent: for instance, the time to peak of DTDUHs presented on figures 6 and 8 do not vary with the saturation level

alpha, which does not seem consistent with eq.9; also, the DTDUHs on figures 6 and 8 show areas under curves which largely vary with the level of saturation alpha of the basins, whereas on figures 9 and 10 the areas under curves do not vary anymore with the saturation level.

**Response:**

Sorry for being unclear about the method, and we will clarify it more clearly in the revised manuscript. We hope the responses bellow could address your question.

(1) For your first question, based on Eq. (9), the flow velocity varies with the state of the soil moisture in unsaturated areas ($\theta_t$) for the TDUH, and the time to peak of TDUHs vary with each other. However, we extracted the saturated areas based on the TWI for the DTDUHs, and the derived unit hydrographs correspond only to the saturated areas. To that end, $\theta_t$ is almost equal to 1 when deriving DTDUHs, and Eq. (9) turned to be Eq. (10), which can be given by

$$V = k \cdot S^{1/2} \cdot \left(\frac{I_t}{I_c}\right)^{2/5} \cdot (\theta_t)^{\gamma} \tag{9}$$

$$V = k \cdot S^{1/2} \cdot \left(\frac{I_t}{I_c}\right)^{2/5} \tag{10}$$

Therefore, the time to peak varies not very significant with the soil moisture.

(2) For your second question, as we can see from Fig. 6 and 8 in the original manuscript, the shape of DTDUHs significantly vary from different soil moisture. This is because the definition of the DTDUH is that a typical hydrograph of direct runoff which gets generated from one centimetre of effective rainfall falling at a uniform rate over the saturated drainage basin uniformly during a specific duration, which means that we computed the DTDUH corresponding to the runoff generating area. While the depths of the surface runoff calculated by Eq. (4) corresponds to the whole basin. To this end, when the DTDUH was adopted as the runoff routing method, the depths of the surface runoff should be converted from the whole basin to the saturated area based on Eq. (11).

$$RS_s^{'} = \frac{RS_s}{\alpha_t} \tag{11}$$

where $\alpha_t$ is the proportion of the saturated area over the whole basin.

For example, assume the state of soil moisture is 0.5 at time $t$ ( $\alpha_t = 0.5$ ), and when there accrues 10 mm excess-rainfall over the whole basin, which means that there will be 20 mm excess-rainfall generated over the saturated areas for the DTDUH method, while there will be only 10 mm excess-rainfall over the whole basin for the traditional TDUH method. On such conditions, although the shape of DTDUHs vary with each other for different soil moisture state, the flow hydrograph at the outlet of the watershed has very small differences.

Considering all these weaknesses, I would not recommend to publish this paper in its current form.

**Detailed comments**

**Comment 1:**- l. 10   rather « a common challenging issue in ..”

Thank you for your comments. “A common issue in challenging hydrological modelling” will be changed as “A common challenging issue in hydrological modelling”.

**Comment 2:**- l. 25 “as for the TDUH method”

Thank you for your comments. “as is the same with the TDUH method” will be changed as “as for the TDUH method”.

**Comment 3:**- l. 48 specify here “the watershed response to efficient rainfall”, since the response to rainfall (including the transformation to efficient rainfall) is obviously not linear..

Thank you for your comments. This sentence will be corrected as “The UH method assumes the watershed response to efficient rainfall be linear and time invariant, and rainfall to be spatially homogeneous”.

**Comment 4:**- l. 55 “high intensity of rainfall”

Thank you for your comments. It will be corrected in the revised manuscript.

**Comment 5:**- l. 66 “have attracted much attention”

Thank you for your comments. It will be corrected in the revised manuscript.

**Comment 6:**- l. 83 “the approximations”

Thank you for your comments. It will be corrected in the revised manuscript.

**Comment 7:**- l. 93 “This raised the question whether ”

Thank you for your comments. It will be changed in the revised manuscript.

**Comment 8:**- l. 103-105 I think a reference to the work of Andrieu et al. (2021), who

proposed e-GUIHs accounting for the spatial variability of rainfall (and thus indirectly the spatial variability of efficient rainfall) would be relevant here.

Thank you for your comments. We have carefully read the article by Andrieu et al. (2021), and this research is closely related to our study. It will be added in the revised manuscript.

**References:**

Andrieu, H., Moussa, R., Kirstetter, P.-E., 2021. The Event-specific Geomorphological Instantaneous Unit Hydrograph (E-GIUH): The basin hydrological response characteristic of a flood event. Journal of Hydrology, 603: 127158. https://doi.org/10.1016/j.jhydrol.2021.127158.

**Comment 9:**- l.110 rather "the runoff generating areas"

Thank you for your comments. "runoff generated areas" will be corrected as "runoff generating areas".

**Comment 10:**- l.111-112 I have some difficulties with this sentence: what do you mean by "unify the spatial scales of the runoff generation and the confluence method" ? Please reformulate

Thank you for your comments. We revised this sentence as "The XAJ model was used as the hydrological modelling framework to compare the performances of TDUH and DTDUH based on flood events".

**Comment 11:**- l. 113 "Finally, .. "Please remove the word finally since the case studies are directly linked to the questions and evaluation presented before.

Thank you for your comments. We will delete this word in the revised manuscript.

**Comment 12:**- l.115-116 A short presentation of the structure of the paper would be good here.

Thank you for your suggestion. The added sentences are as follows:

"The remaining chapters of this paper are arranged as follows: In Hydrological models, the processes of DTDUH considering the spatial heterogeneity of runoff generation are introduced, and the parameter calibration method, evaluation criteria as well as the hydrologic model are demonstrated. In section study area and data, the rainfall and runoff data and the study areas are described. In section results, the performances and results of the TDUH and DTUHD are compared. In section discussion, the results and methods were discussed and in section conclusions, relevant conclusions are drawn."

**Comment 13:**- l.118-120 Please moderate this statement, since subsurface flows can also be a significant source of efficient rainfall, when infiltration capacities are larger than rainfall intensities.

Thank you for your comments. We corrected it as "Mostly, saturation-excess runoff is the major runoff mechanism in humid areas."

**Comment 14:-** l.128 – 153 This description of the production part of the hydrological model is not clear. Please define more clearly each notion and variable (Aps, Ap, Af and As, free water storage and tension water storage, AU, FR, BU, S, SM, ..), and the way they are used and related, if necessary by including additional figures. Figure 2 could be placed here, but it does not provide a sufficient level of detail to understand how the model's variables and parameters are related. Finally, I do not understand how the areas Aps, Ap, Af, As are used in the computation, and how eq. 3 and equations 4, 5, 6   can be combined to ensure a conservation of volume (I guess R should be equal to RS+RI+RG but this does not seem consistent with the equations)

We are sorry for the unclear description of the modeling process.

① We will add a diagram of the free water storage and tension water storage (Fig. 1).

② We will define each notion and variable more clearly.

③ Aps, Ap, Af, As are used in Eqs. (2) and (4), to calculate the total runoff and the surface runoff, interflow and groundwater.

④ $R$ is equal to RS+RI+RG, and it will be added in the revised manuscript.

We will reorganize this part, and the revised version is as follows:

[Figure]

**Figure 1.** (a) Tension water storage capacity curve, and (b) Free water storage capacity curve

The difference between field capacity and total soil water content is defined as tension water storage capacity, the maximum amount of water available in unsaturated zone. In the XAJ model, a tension water capacity curve is introduced (Fig. 1(a)) to describe the non-uniform distribution of tension water capacity throughout basin or sub-basin. In Fig. 1(a), $A_{ps}$ is the partial pervious area where the tension water storage capacity is less than or equal to the value $W$, which is the tension water capacity at a point, varying from 0 to a maximum $WMM$; $A_p$ is the pervious area; $W_0$ is the initial areal mean tension water storage (mm); and $AU$ is the vertical coordinate corresponding to $W_0$. The functional relationship of the tension water storage capacity curve is expressed as:

$$\frac{A_{ps}}{A_p} = 1 - \left(1 - \frac{W}{WMM}\right)^B \tag{1}$$

Based on Eq. (1), when rainfall exceeds evaporation, the runoff generated in the saturated areas can be calculated as:

$$R = \begin{cases} \int_{AU}^{AU+PE} \frac{A_{ps}}{A_p} \, dW & PE + AU < WMM \\ PE - WM + W_0 & PE + AU \geq WMM \end{cases} \tag{2}$$

The total runoff $R$ ($R=RS+RI+RG$), generated in a wet period in accordance with Eq. (2), must be separated into three components, including the surface runoff, the subsurface stormflow and the surface runoff. Thus, the concept of free water storage capacity was used, and it was assumed to be distributed between zero and a point maximum $SMM$ in a parabolic manner, as shown in Fig. 1(b). In Fig. 1(b), $A_p$ is the pervious area of the catchment; $A_f$ is the area where the free water storage capacity is less than or equal to the value $S$, varying from 0 to $SMM$; $A_s$ is the runoff generation area; $S_0$ is the initial areal mean free water storage (mm); $BU$ is the vertical coordinate corresponding to $S_0$; and $RI$, $RG$ represent the depth of the interflow and subsurface flow. The functional relationship of the free water storage capacity curve is expressed as:

$$\frac{A_f}{A_s} = 1 - \left(1 - \frac{S}{SMM}\right)^{EX} \tag{3}$$

The total runoff $R$, generated in a wet period in accordance with Eq. (2), can be subsequently separated into three components, including the surface runoff, interflow and groundwater, which can be given by

$$RS = \begin{cases} FR \int_{BU}^{BU+PE} \frac{A_f}{A_s} \, dS & PE + BU < SMM \\ FR\left(PE + S_0 - SM\right) & PE + BU \geq SMM \end{cases} \tag{4}$$

$$RI = KI \cdot FR \cdot \int_0^{BU} \frac{A_f}{A_s} \, dS \tag{5}$$

$$RG = KG \cdot FR \cdot \int_0^{BU} \frac{A_f}{A_s} \, dS \tag{6}$$

where $RS$, $RI$, $RG$ represent the depth of the surface runoff, interflow and groundwater respectively (mm); $FR$, equaling to $R/PE$, is the proportion of the runoff producing area over the whole basin; $SM$ is the areal mean free water capacity (mm); and $KI$ and $KG$ are outflow coefficients of the free water storage to interflow and groundwater, respectively.

**Comment 15:**- l. 173 "the total number of grid cells"

Thank you for your comments. It will be corrected in the revised manuscript.

**Comment 16:**- l.175 "The depth of the excess rainfall occurs only in the saturated areas

when the entire basin does not reach a global saturated state": this appears as a theory that is only rarely valid.

Thank you for your comments. Sorry for being unclear on this. The Xinanjiang (XAJ) model (Zhao, 1992) is a conceptual lumped hydrological model that has a wide range of applications in China. The key concept of the model is the saturation excess runoff generation mechanism, that is, there is no runoff generated until the tension water capacity is satisfied. We will revise this sentence as:

"The XAJ model adopted the saturation excess runoff generation mechanism, that is, there is no runoff generated until the tension water capacity is satisfied."

**References:**

Zhao, R. J, 1992. The Xinanjiang model applied in China. Journal of hydrology 135.1-4: 371-381. https://doi.org/10.1016/0022-1694(92)90096-E.

Zhao, J. F., Duan, Y., Hu, Y., Li, B., & Liang, Z, 2023. The numerical error of the Xinanjiang model. Journal of Hydrology, 619: 129324. https://doi.org/10.1016/j.jhydrol.2023.129324.

**Comment 17:**- l.186 the presence of the alpha (soil moisture) variable in eq.9 corresponds to the assumption that velocities vary with the global soil moisture. This also suggests that flow generation does not only correspond to saturation excess.

Thank you for your comments. We adopted a new equation (Eq. 9) of the flow velocity proposed in our previous research. Current method assumed that equilibrium in each individual grid cell was reached before the end of the rainfall excess pulse. When continuous excess rainfall accrues in a watershed, the soil moisture content and surface runoff increase, and the infiltration rate decreases, leading to an acceleration of flow routing velocity, until the entire basin is saturated, and the routing velocity reaches its maximum. This assumption of equilibrium globally or in grid cells yields faster travel flow velocities, smaller travel time and higher peak discharge. To that end, a soil moisture factor $\theta_t$ was introduced to characterize the soil moisture content in unsaturated areas.

Simultaneously, we agree with the reviewer that flow generation does not only correspond to saturation excess, as the infiltration-excess is also important. Therefore, the proposed method is limited in the humid or semi-humid areas, which is dominated by saturation excess.

**Comment 18:**- l.213-215 "To compare the differences …" Again here I feel this sentence not very clear, please could you reformulate this.

Sorry for being unclear on this. It will be revised as: "To investigate the influence of spatial heterogeneity of runoff generation on the runoff routing, linear reservoir, TDUH and DTDUH were selected as the surface runoff routing methods."

**Comment 19:**- l.215-216 I do not see any application of the Muskingum method in the presented routing schemes.

Sorry for being unclear on this. We want to express that when the watershed is divided into multiple sub-basins, the Muskingum method will be used to confluence the runoff of each sub-basin to the outlet of the basin. It will be reformulated in the revised manuscript.

**Comment 20:** Figure 2. According to this figure and the text, it seems that the routing schemes are applied only on RS (surface runoff). Could you explain what happens with subsurface and groundwater components (RI and RG): are they also routed and how?

Thank you for your comments. Fig. 2 presented the schematic diagram of the XAJ model. For the runoff routing module in the XAJ model, the subsurface stormflow (*RI*) and subsurface runoff (*RG*) were considered using a free reservoir, but the proposed routing schemes are applied only on RS. Their expressions are given by

$$QI_t = CI \cdot QI_{t-1} + (1 - CI) \cdot RI_t \cdot U \tag{7}$$

$$QG_t = CG \cdot QG_{t-1} + (1 - CG) \cdot RG_t \cdot U \tag{8}$$

[Figure]

**Figure 2.** Schematic diagram of the XAJ model

**Comment 21:** - l.224 – 230 Again here, it is suggested that the discharge at the basin outlet is computed by routing only the surface runoff and is resumed to QS. Can you at least explain if Qi and QG are neglected and if the validity of this assumption has been verified for the presented case studies?

Thank you for your comments.

First, the discharge at the basin outlet is computed by routing all the three runoff components, including RS, RI and RG. The surface runoff (RS) was routed using the unit hydrograph, and the subsurface stormflow (RI) and subsurface runoff (RG) were considered using a free reservoir.

Second, in Section 4.4 and 4.5 of the original manuscript, we have clarified the rationality of this routing scheme, and we will reorganize it in the revised manuscript as follows:

The performances of the model were evaluated based on the flow hydrograph at the outlet of the watershed, and the flow hydrograph was composed by three components including the

surface runoff, subsurface stormflow runoff and subsurface runoff. Since the TDUH and the DTDUH were used for the surface runoff routing, and the subsurface stormflow and subsurface runoff were considered using a free reservoir. To evaluate the performances of the TDUH and DTDUH under the condition of XAJ modelling framework, we are supposed to calculate the ratio of surface runoff to the total depth of excess rainfall. When the surface runoff constitutes the majority of the total runoff, the study area was considered rational. Figs. 3(a) and 3(b) show that the surface runoff accounts for most of the total depth of excess rainfall, which means that the performances of the hydrological model are mainly affected by the surface runoff routing methods. To this end, it is reasonable to compare the performances of the TDUH and DTDUH in the Longhu and Dongshi River basins.

[Figure]

**Figure 3.** Details of the runoff components of the 16 flood events for the (a) Longhu River basin. (b) Dongshi River basin.

**Comment 22:**- l.237 – 238 "aimed at maximizing flow characteristics": not clear , please reformulate.

Thank you for your comments. It will be revised as "An aggregated objective function made up of three measures was used for the parameter calibration (Brunner et al., 2021)"

**References:**

Brunner, M. I., Swain, D. L., Wood, R. R., Willkofer, F., Done, J. M., Gilleland, E., Ludwig, R., 2021. An extremeness threshold determines the regional response of floods to changes in rainfall extremes. Communications Earth & Environment, 2(1): 173. https://doi.org/10.1038/s43247-021-00248-x.

**Comment 23:**- l.248 – 255 Could you justify here the choice of different parameters for evaluation and for calibration?

We appreciate for this comment. Different parameters for evaluation and for calibration will be clarified here. A table will be added in the revised manuscript:

**Table 1**. Explanation of different parameters for evaluation and for calibration.

| Description | Notation | Explanation |
|---|---|---|
| Ratio of potential evapotranspiration to pan evaporation | $KC$ (unitless) | |
| Averaged soil moisture storage capacity of the upper layer | $UM$ (mm) | |
| Averaged soil moisture storage capacity of the lower layer | $LM$ (mm) | |
| Averaged soil moisture storage capacity of the deep layer | $DM$ (mm) | |
| Exponential of the distribution to tension water capacity | $B$ (unitless) | 15 parameters of the XAJ model, and these parameters are calibrated based on the SCE-UA method. (*KE and XE are required when a watershed was divided into several sub-basins*) |
| Percentage of impervious in the watershed | $IM$ (unitless) | |
| Evapotranspiration coefficient of the deeper layer | $C$ (unitless) | |
| Mean free water capacity of the surface soil layer | $SM$ (mm) | |
| Exponent of the distribution to free water capacity | $EX$ (unitless) | |
| Outflow coefficients of the free water storage to subsurface stormflow | $KI$ (unitless) | |
| Outflow coefficients of the free water storage to subsurface flow | $KG$ (unitless) | |
| Recession constants of the subsurface stormflow | $CI$ (unitless) | |
| Recession constants of the surface runoff storage | $CG$ (unitless) | |
| Recession constants of channel network storage | $CS$ (unitless) | |
| Muskingum time constant | $KE$ (h) | |
| Muskingum weighting factor | $XE$ (unitless) | |
| Lag in time | $L$ (h) | |
| Reference rainfall intensity | $I_c$ (mm/h) | Evaluated based on the average rainfall intensity |
| Power law related to the influence of soil moisture on flow velocity | $\gamma$ (unitless) | Evaluated based on the trial way, and it is usually lower than 0.7 (Yi et al., 2022) |
| Slope of the watershed grid cell | $S$ (m/m) | Evaluated based on DEM of the watershed |
| Land use or flow type coefficient | $K$ (m/s) | Evaluated based on different underlying surface types or different flow states (Ajward and Muzik, 2000) |

**References:**

Yi, B., Chen, L., Zhang, H., Singh, V. P., Jiang, P., Liu, Y., Guo, H., Qiu, H., 2022. A time-varying distributed unit hydrograph method considering soil moisture. Hydrology and Earth System Sciences, 26(20): 5269-5289. https://doi.org/10.5194/hess-26-5269-2022.

Ajward, M. H., Muzik, I., 2000. A spatially varied unit hydrograph model. Journal of Environmental Hydrology, 8(7).

**Comment 24:**- l.262 – 263 Please mention here the names of institutions providing data, in addition to URLs.

Thank you for your comments. The DEM data was obtained from Geospatial Data Cloud (https://www.gscloud.cn/). The land cover data was collected from Tsinghua University (http://data.ess.tsinghua.edu.cn/).

**Comment 25:-** l.269 – 271 Here again, could you mention the data providers for rainfall and discharge data?

Thank you for your comments. Meizhou Hydrological Bureau provides the rainfall and runoff data. It will be added in the revised manuscript.

**Comment 26:-** l.271 – 272 Do you mean here that the model was calibrated only on flood periods?

Thank you for your comments. A total of 16 isolated storms were identified from the continuous flow process in the Longhu and Dongshi River basins, respectively. The model was calibrated based on these flood events.

**Comment 27:-** l.271 – 272  and table 1 Could you mention which events were used for calibration and which ones for verification?

Thank you for your comments. 12 flood events were used to calibrate the model, and 4 flood evets were used for verification, respectively, for the two basins. The calibrated and verified flood events will be clarified in the revised manuscript. The revised table is as follows:

**Table 2**. Statistics of the flood events in the Longhu and the Dongshi River basins.

| Watershed | Periods | Flood events | Rainfall (mm) | Flood peak ($m^3 \ s^{-1}$) | Time duration (h) |
|---|---|---|---|---|---|
| Longhu | Calibration | 19730508 | 80.0 | 94.5 | 27 |
| | | 19730720 | 76.7 | 180.0 | 17 |
| | | 19750526 | 54.9 | 101.0 | 21 |
| | | 19760702 | 73.0 | 137.0 | 28 |
| | | 19770526 | 73.8 | 90.4 | 18 |
| | | 19771003 | 62.1 | 97.5 | 19 |
| | | 19790607 | 100.3 | 93.4 | 24 |
| | | 19890502 | 46.5 | 132.0 | 29 |
| | | 20030517 | 94.0 | 140.0 | 46 |
| | | 20120527 | 56.0 | 96.5 | 37 |
| | | 20130713 | 118.8 | 128.0 | 27 |
| | | 20150601 | 214.4 | 228.0 | 30 |
| | Verification | 20150831 | 83.4 | 85.0 | 44 |
| | | 20160430 | 102.6 | 83.2 | 30 |
| | | 20160903 | 111.2 | 91.0 | 54 |
| | | 20161021 | 85.4 | 89.7 | 26 |

| Watershed | Periods | Flood events | Rainfall (mm) | Flood peak ($m^3$ $s^{-1}$) | Time duration (h) |
|-----------|---------|--------------|---------------|-----------------------------|--------------------|
| Dongshi | Calibration | 20150509 | 105.2 | 62.9 | 38 |
| | | 20150721 | 132.0 | 82.0 | 29 |
| | | 20160811 | 90.0 | 51.3 | 48 |
| | | 20160819 | 112.5 | 34.9 | 19 |
| | | 20161021 | 158.8 | 48.0 | 49 |
| | | 20170501 | 84.5 | 98.3 | 22 |
| | | 20170515 | 84.0 | 43.7 | 29 |
| | | 20170613 | 139.2 | 37.2 | 31 |
| | | 20170929 | 71.0 | 101.2 | 25 |
| | | 20180606 | 61.5 | 34.9 | 32 |
| | | 20180702 | 23.5 | 44.3 | 25 |
| | | 20190418 | 86.4 | 35.5 | 18 |
| | Verification | 20190609 | 107.6 | 272.0 | 27 |
| | | 20190612 | 74.0 | 100.0 | 66 |
| | | 20200522 | 67.5 | 71.0 | 37 |
| | | 20200607 | 109.3 | 50.6 | 26 |

**Comment 28:**- l.295 – 296 It is rather surprising here that the routing parameters Ic and gamma do not vary from one event to another, and seem to have been calibrated formerly, i.e. independently of the calibration of the production part of the model. Could you better justify this choice?

Thank you for your comments.

The flow velocity formula was proposed in our previous study (Yi et al., 2022), and the objective of this study was to explore the influence of spatial heterogeneity of runoff generation on the distributed unit hydrograph for flood prediction. parameter $I_c$ represents the reference rainfall intensity, and could be evaluated based on the average rainfall intensity. Simultaneously, $\gamma$ is a power law related to the influence of soil moisture on flow velocity, and the sensitivity analysis for variable gamma has been made in our previous study. The results shown that the mean flow velocity of the basin was significantly influenced by exponent $\gamma$. In addition, when the soil moisture content exceeded 0.7, the variation range of mean flow velocity decreased sharply, which indicated that the influence of parameter $\gamma$ on the flow velocity decreased gradually with the increase of soil moisture content.

In theory, $I_c$ and $\gamma$ are different from one flood event to another. However, it is difficult to realize in practical use, and we usually adopted the unit hydrograph charactering the average physical properties of a watershed. Thus, in practical flood forecasting, the parameter $\gamma$ should be a constant once it was determined. This is similar with the influence of upstream contributions to the flow velocity formula in previous research (Leopold & Miller, 1956; Rodríguez-Iturbe et al., 1992; Rodriguez-Iturbe & Rinaldo, 1997; Leopold et al., 2012; Bhattacharya et al., 2012).

**References:**

Bhattacharya, A. K., McEnroe, B. M., Zhao, H., Kumar, D., & Shinde, S. Modclark model: Improvement and application. International Journal of Civil Engineering, 2(7), 100- 118, 2012.

Leopold, L. B., & Miller, J. P. Ephemeral Streams: Hydraulic Factors and their Relation to the Drainage Net (Vol. 282). Arlington, VA: US Government Printing Office, 1956.

Leopold, L. B., Wolman, M. G., & Miller, J. P. Fluvial Processes in Geomorphology. Mineola, New York: Courier Corporation, 2012.

Rodríguez-Iturbe, I., Ijjász-Vásquez, E. J., Bras, R. L., & Tarboton, D. G. Power law distributions of discharge mass and energy in river basins. Water Resources Research, 28(4), 1089– 1093. https://doi.org/10.1029/91WR03033, 1992.

Rodriguez-Iturbe, I., & Rinaldo, A. Fractal River Basins: Chance and Self-Organization. Cambridge, UK: Cambridge University Press, 1997.

**Comment 29:**- l.297 – 297 This statement is very important to understand the methodological choices. This should have been mentioned at the beginning of this paper.

Thank you for your comments. This statement will be mentioned in the abstract of the revised version.

**Comment 30:**- l.300 – 301 "and thus the parameters of the runoff generation module kept unchanged" This statement is not consistent with explanations provided at lines 306-316, which suggest that the models were calibrated separately. Maybe it would be better to include this statement in lines 317-324 where the explanation is provided.

Thank you for your comments. We have calibrated the three models based on your comments, and this section will be revised accordingly.

**Comment 31:**- l. 306-324 The methodological choice to keep only the XAJ+LR calibration results, for the evaluation of the two other models (XAJ+TDUH and XAJ+DTDUH) seems very surprising to me, since the calibration results resented in appendix suggest that these two model may perform similarly to the XAJ+LR model, when appropriately calibrated. Moreover, the choice to calibrate the models on the whole dataset (without preserving validation data) is also curious. I think it would have been more relevant to calibrate the three models, and to preserve a validation datset and/or to provide cross validation results.

Thank you for your comments. We have calibrated the three models (XAJ+LR, XAJ+TDUH and XAJ+DTDUH), respectively, and these results will be added in the revised manuscript.

Table 3 lists the *NSE*, *RMSE*, error of flood peak ($Q_p$) and error of time to peak ($T_p$) values for all the models of the Longhu River basin, among which No. 20150831, 20160430, 20160903 and 201651021 are validation dataset. To demonstrate the model performances of different strategies more visually, Fig. 4 shows line charts of the three runoff routing methods for the four indexes in both calibration and validation periods. Results shown that TDUH and DTDUH have consistent performances in *NSE*, *RMSE* and $T_p$, and results of the LR method are not stable, sometimes better than that of the DTDUH method, sometimes worse (especially for the criteria $Q_p$). In the calibration periods, DTDUH performed better than that of the TDUH, while in the validation periods, TDUH and DTDUH performed almost consistent. It may be due to the initial conditions of flood events are different from each other. For the calibration periods, the average *NSE* of the LR, TDUH, DTDUH methods are 0.58, 0.61 and 0.64, respectively, and 0.80, 0.78 and 0.82 for the validation periods. Simultaneously, the absolute error of the flood peak are 0.36, 0.25 and 0.24 for the calibration periods, respectively, and 0.18, 0.06 and 0.03 for the validation periods. In general, the improvement from the DTDUH is small, and we summarized the possible reasons as follows 1) The surface runoff accounts for 60% ~ 70% of the total runoff, and it is necessary to consider the influence of the heterogeneity of the subsurface stormflow as well as the subsurface runoff; 2) the antecedent soil moisture is high, and, thus, the simulation error caused by the spatial heterogeneity of runoff generation is small.

**Table 3**. Calibrated and Validated results of the Longhu River basin.

| Flood events | *NSE* | | | *RMSE* (m³/s) | | | $Q_p$ | | | $T_p$ (h) | | |
|---|---|---|---|---|---|---|---|---|---|---|---|---|
| | LR | TDUH | DTDUH | LR | TDUH | DTDUH | LR | TDUH | DTDUH | LR | TDUH | DTDUH |
| 19730508 | 0.77 | 0.83 | 0.85 | 13.66 | 11.67 | 11.11 | -0.28 | -0.22 | -0.21 | 3 | 3 | 3 |
| 19730720 | 0.84 | 0.87 | 0.90 | 22.09 | 19.85 | 17.69 | -0.22 | -0.05 | -0.05 | 1 | 0 | 0 |
| 19750526 | 0.60 | 0.76 | 0.86 | 17.04 | 13.24 | 10.13 | -0.36 | -0.27 | 0.18 | 0 | -1 | 0 |
| 19760702 | 0.59 | 0.68 | 0.67 | 21.96 | 19.45 | 19.73 | -0.53 | -0.39 | -0.39 | 1 | 3 | 3 |
| 19770526 | 0.38 | 0.11 | 0.10 | 21.83 | 26.09 | 26.28 | -0.30 | -0.19 | -0.19 | 3 | 2 | 2 |
| 19771003 | 0.54 | 0.32 | 0.37 | 19.29 | 23.47 | 22.60 | -0.28 | -0.20 | -0.19 | 0 | 3 | 2 |
| 19790607 | 0.31 | 0.34 | 0.47 | 23.41 | 22.91 | 20.39 | -0.28 | -0.25 | -0.25 | 2 | 2 | 2 |
| 19890502 | 0.46 | 0.49 | 0.52 | 28.24 | 27.65 | 26.80 | -0.64 | -0.66 | -0.50 | -1 | 1 | 0 |
| 20030517 | 0.49 | 0.54 | 0.54 | 31.63 | 30.30 | 30.07 | -0.35 | -0.32 | -0.32 | 0 | -1 | -1 |
| 20120527 | 0.72 | 0.73 | 0.72 | 21.15 | 20.69 | 21.03 | -0.25 | 0.03 | 0.03 | -2 | 0 | 0 |
| 20130713 | 0.70 | 0.82 | 0.81 | 37.50 | 28.74 | 30.22 | -0.33 | -0.30 | -0.32 | 1 | 0 | 0 |
| 20150601 | 0.52 | 0.84 | 0.82 | 19.08 | 10.94 | 11.70 | 0.50 | 0.15 | 0.25 | 0 | -1 | -1 |
| 20150831 | 0.82 | 0.74 | 0.76 | 11.88 | 14.12 | 13.58 | -0.21 | -0.02 | 0.00 | -3 | 1 | 1 |
| 20160430 | 0.93 | 0.72 | 0.77 | 7.33 | 13.29 | 13.12 | 0.01 | 0.11 | 0.08 | 1 | 1 | 1 |
| 20160903 | 0.70 | 0.76 | 0.81 | 16.67 | 13.20 | 13.75 | -0.36 | -0.09 | -0.04 | -4 | 0 | 0 |
| 20161021 | 0.73 | 0.90 | 0.94 | 17.22 | 8.47 | 8.50 | -0.16 | 0.03 | -0.01 | -2 | -1 | -1 |

[Figure]

**Figure 4.** Line chart of the *NSE*, *RMSE*, $Q_p$ and $T_p$ for the Longhu River basin.

Simultaneously, Table 4 lists the *NSE*, *RMSE*, error of flood peak ($Q_p$) and error of time to peak ($T_p$) values for all the models of the Dongshi River basin, among which No. 20190609, 20190612, 20200522 and 20200607 are validation dataset. To demonstrate the model performances of different strategies more visually, Fig. 5 shows line charts of the three runoff routing methods for the four indexes in both calibration and validation periods. Compared with the results of Longhu River, the DTDUH and LR methods shown consistent performances, and significantly better than that of the TDUH method. It shows that the DTDUH method shows significant improvement in this basin.

**Table 4**. Calibrated and validated results of the Dongshi River basin.

| Flood events | *NSE* | | | *RMSE* (m³/s) | | | $Q_p$ | | | $T_p$ (h) | | |
|---|---|---|---|---|---|---|---|---|---|---|---|---|
| | LR | TDUH | DTDUH | LR | TDUH | DTDUH | LR | TDUH | DTDUH | LR | TDUH | DTDUH |
| 20150509 | 0.37 | 0.09 | 0.32 | 12.61 | 15.17 | 13.08 | -0.73 | -0.83 | -0.76 | 5 | 5 | 5 |
| 20150721 | 0.81 | 0.87 | 0.80 | 11.21 | 9.25 | 11.60 | 0.02 | -0.16 | 0.06 | -1 | -1 | -1 |
| 20160811 | 0.61 | 0.54 | 0.67 | 5.17 | 5.63 | 4.78 | -0.34 | -0.17 | -0.38 | 1 | 1 | 1 |
| 20160819 | 0.83 | 0.68 | 0.79 | 2.39 | 3.34 | 2.71 | -0.26 | -0.40 | -0.26 | 0 | 0 | 0 |
| 20161021 | 0.93 | 0.81 | 0.94 | 6.99 | 11.80 | 6.75 | -0.11 | -0.27 | -0.10 | 2 | 5 | 2 |
| 20170501 | 0.89 | 0.67 | 0.86 | 2.69 | 4.75 | 3.11 | -0.16 | -0.27 | 0.00 | 0 | 1 | 0 |
| 20170515 | 0.64 | 0.78 | 0.64 | 3.95 | 3.10 | 3.97 | -0.05 | -0.14 | -0.07 | 0 | 4 | 4 |
| 20170613 | 0.91 | 0.76 | 0.91 | 8.40 | 13.48 | 8.28 | -0.26 | -0.38 | -0.22 | 2 | 2 | 2 |
| 20170929 | 0.75 | 0.53 | 0.73 | 3.71 | 5.12 | 3.87 | -0.20 | -0.21 | 0.17 | -3 | 2 | 0 |
| 20180606 | 0.55 | 0.34 | 0.53 | 5.53 | 6.73 | 5.68 | -0.51 | -0.55 | -0.55 | 1 | 1 | 1 |
| 20180702 | 0.55 | 0.59 | 0.48 | 4.17 | 3.96 | 4.48 | -0.51 | -0.45 | -0.58 | 2 | 2 | 2 |

| | | | | | | | | | | | | |
|---|---|---|---|---|---|---|---|---|---|---|---|---|
| 20190418 | 0.61 | 0.55 | 0.58 | 7.54 | 8.00 | 7.79 | -0.63 | -0.67 | -0.65 | 2 | 12 | 2 |
| 20190609 | 0.55 | 0.56 | 0.75 | 13.27 | 13.03 | 9.83 | -0.54 | -0.53 | -0.42 | 2 | 2 | 2 |
| 20190612 | 0.81 | 0.76 | 0.83 | 4.57 | 5.09 | 4.36 | -0.18 | -0.06 | -0.02 | 0 | 0 | 0 |
| 20200522 | 0.77 | 0.56 | 0.75 | 5.24 | 7.32 | 5.56 | -0.01 | -0.02 | -0.04 | 2 | 2 | 2 |
| 20200607 | 0.57 | 0.54 | 0.61 | 28.85 | 30.03 | 27.54 | -0.50 | -0.54 | -0.50 | 0 | -1 | 2 |

[Figure]

**Figure 5.** Line chart of the *NSE*, *RMSE*, $Q_p$ and $T_p$ for the Dongshi River basin.

**Comment 32:**- l.337-347 Could you explain here how do you retrieve the value of alpha_t based on the structure of the model presented in figure 2? Could you also mention how the value of alpha_t is converted to a saturated surface: assumption that alpha_t=x corresponds to a x% of saturated cells in the watershed (corresponding to the x% cells having the largest TWI values)?

First, sorry for being unclear here. We will correct $\alpha_t$ as $\theta_t$ in the revised manuscript.

$\alpha_t$ represents the saturated area at time t, and $\theta_t$ represents the state of the soil moisture content of the unsaturated areas. The formula of the flow velocity was given by Eq. (9). We think you want to know how we can retrieve the value of $\theta_t$ in Eq. (9).

$$V = k \cdot S^{1/2} \cdot \left( \frac{I_t}{I_c} \right)^{2/5} \cdot \left( \theta_t \right)^{\gamma} \tag{9}$$

Fig. 6 shows the tension water storage capacity curve in the XAJ model, and the specific formula (Moore, 1985) is given by Eq. (10)

[Figure]

**Figure 6.** Tension water storage capacity curve

$$\alpha = 1 - \left(1 - \frac{WM}{WMM}\right)^{B}$$ (10)

where $\alpha$ $\left(\alpha = \dfrac{A_{ps}}{A_p}\right.$, unitless) represents the proportion of the pervious area of the basin whose tension water capacity is less than or equal to the value of the ordinate $WM$ (m); the tension water capacity at a point, $WM$ varies from 0 to $WMM$; $WMM$ (m) is maximum watershed soil storage capacity; $B$ represents the degree of spatial variability of store capacity over the basin; and the area under the curve represents the areal mean tension capacity of the entire basin.

The state of the catchment at any time $t$, can be represented by a point $x(\alpha_t, WM_t)$ on the curved line of Figure 4 (Zhao, 1992), which implies

$$\alpha_t = 1 - \left(1 - \frac{WM_t}{WMM}\right)^{B}$$ (11)

The area to the right and below the point $x$ is proportional to the areal mean tension water storage (not capacity). Thus, $WM_t$ (m) the ordinate of the point $x$ represents the tension water storage capacity in the basin at time $t$; $w_t$ (m) can be assumed to represent the mean tension

water storage of the unsaturated region, and $w_{max,t}$ (m) represents the maximum tension water storage of the unsaturated region at time $t$. The expressions are given by

$$w_t = \left(1-\alpha_t\right) \cdot WM_t \tag{12}$$

$$w_{max,t} = \int_{\alpha_t}^{1} WMM \left[1-\left(1-\alpha\right)^{\frac{1}{B}}\right] d\alpha \tag{13}$$

The state of the soil moisture content $\theta_t$ of the unsaturated areas is the ratio $w_t$ and $w_{max,t}$, which implies

$$\theta_t = \frac{w_t}{w_{max,t}} = \frac{\left(1-\alpha_t\right)\cdot WM_t}{\int_{\alpha_t}^{1} WMM \left[1-\left(1-\alpha\right)^{\frac{1}{B}}\right] d\alpha} \tag{14}$$

Substitute Eq. (11) to Eq. (14), which yields

$$\theta_t = \frac{\left(1-\alpha_t\right)\cdot WM_t}{WMM \left[1-\alpha_t - \frac{B}{B+1}\left(1-\alpha_t\right)^{1+\frac{1}{B}}\right]} = \frac{\left(B+1\right) WM_t}{WMM + BWM_t} \tag{15}$$

where $\theta_t$ (unitless) represents the state of the soil moisture content of the unsaturated areas at time $t$. More details can be found in Yi et al. (2022).

Second, when $\alpha_t = x$, it corresponds to a $x\%$ of saturated cells in the watershed, and also corresponds to the $x\%$ cells having the largest TWI values.

**References:**

Moore, R. J.: The probability-distributed principle and runoff production at point and basin scales, Hydrol. Sci. J., 30, 273–297, 1985.

Yi, B., Chen, L., Zhang, H., Singh, V. P., Jiang, P., Liu, Y., Guo, H., Qiu, H., 2022. A time-varying distributed unit hydrograph method considering soil moisture. Hydrology

and Earth System Sciences, 26(20): 5269-5289. https://doi.org/10.5194/hess-26-5269-2022.

**Comment 33:-** l.356-365 and figures 5 to 8: it is rather surprising here that the time to peak of DTDUHs do not vary, depending on alpha_t values, since the eq. 9 used for the computation of velocities still integrates the value of alpha_t. I guess here that for DTDUH the velocities are computed from an equation that differs from eq. 9. Could you clarify this?

Thank you for your comments. As is the same with Comment 32 that $\alpha_t$ should be $\theta_t$.

Based on Eq. (9), the flow velocity varies with the state of the soil moisture in unsaturated areas for the TDUH, and the time to peak of TDUHs vary with each other. However, we extracted the saturated areas based on the TWI for the DTDUHs, and the derived unit hydrographs correspond only to the saturated areas. To that end, $\theta_t$ is almost equal to 1 when deriving DTDUHs, and Eq.(9) turned to be Eq. (16), which can be given by

$$V = k \cdot S^{1/2} \cdot \left(\frac{I_t}{I_c}\right)^{2/5} \tag{16}$$

Therefore, the time to peak varies not very significant with the soil moisture.

**Comment 34:-** Figure 9 and 10 these figures appear inconsistent with figure 6 and 8, since the shape of DTDUHs do not significantly vary here between S1 and S4, whereas large variations in the areas under the curves are observed on figures 6 and 8 (which is justified in lines 360-361 by the fact that DTDUHs are derived only from saturated areas)

Thank you for your comments. As we can see from Fig. 6 and 8 in the original manuscript, the shape of DTDUHs significantly vary from different soil moisture. This is because the definition of the DTDUH is that a typical hydrograph of direct runoff which gets generated from one centimetre of effective rainfall falling at a uniform rate over the saturated drainage basin uniformly during a specific duration, which means that we computed the DTDUH corresponding to the runoff generating area. While the depths of the surface runoff calculated by Eq. (4) corresponds to the whole basin. To this end, when the DTDUH was adopted as the runoff routing method, the depths of the surface runoff should be converted from the whole basin to the saturated area based on Eq. (17). And this is why the shape of DTDUHs do not significantly vary between S1 and S4.

$$RS_s^{'} = \frac{RS_s}{\alpha_t} \tag{17}$$

where $\alpha_t$ is the proportion of the saturated area over the whole basin.

For example, assume the state of soil moisture is 0.5 at time $t$ ($\alpha_t = 0.5$), and when there accrues 10 mm excess-rainfall over the whole basin, which means that there will be 20 mm excess-rainfall generated over the saturated areas for the DTDUH method, while there will be only 10 mm excess-rainfall over the whole basin for the traditional TDUH method. On such conditions, although the shape of DTDUHs vary with each other for different soil moisture state, the flow hydrograph at the outlet of the watershed has very small differences.

We hope the responses above could address your question.

**Comment 35:**- l.404—407 I feel significant differences are observed for both basins, even if more limited for Dongshi basin

Thank you for your comments. We will reformulate the sentences, and the revised sentences are as follows:

"It can also be found that the differences between the results of the TDUH and the DTDUH in the Dongshi River basin are also significant, but shows more limited for the Dongshi River basin than that of the Longhu River basin"

**Comment 36:**- Figure 11 I understand here that the three flow components of the model (surface and two subsurface ones) were kept for the comparison to observed hydrographs. Could you provide information on the routing scheme used for subsurface flows? Could you also provide information about the alpha_t values corresponding to antecedent soil moisture conditions in figure 11.b?

Thank you for your suggestions. We are sorry for being unclear here.

①As presented in comments 20 and 21, we will add routing information of subsurface stormflow and subsurface runoff in the revised manuscript.

②Information about the $\theta_t$ values corresponding to antecedent soil moisture conditions can be calculated based on Eq. (15) for the TDUH method. The $\theta_t$ values corresponding to antecedent soil moisture conditions ($\theta_{t_0}$) in Figs. 11(b) and 13(b) are given in Tables 5. In addition, $\theta_t$ is equal to 1 for DTDUH methods, and this has been discussed in comment 33.

**Table 5.** $\theta_t$ values corresponding to antecedent soil moisture conditions ($\theta_{t_0}$) for the Longhu and Dongshi River basins.

| Longhu | $\theta_{t_0}$ | Dongshi | $\theta_{t_0}$ |
|---|---|---|---|
| 19730508 | 0.62 | 20150509 | 0.32 |
| 19730720 | 0.96 | 20150721 | 1.00 |
| 19750526 | 1.00 | 20160811 | 0.71 |
| 19760702 | 0.92 | 20160819 | 0.71 |
| 19770526 | 0.70 | 20161021 | 1.00 |
| 19771003 | 0.82 | 20170501 | 0.47 |
| 19790607 | 0.35 | 20170515 | 0.55 |
| 19890502 | 0.34 | 20170613 | 1.00 |
| 20030517 | 0.78 | 20170929 | 0.53 |
| 20120527 | 0.76 | 20180606 | 0.76 |
| 20130713 | 0.50 | 20180702 | 0.85 |
| 20150601 | 1.00 | 20190418 | 0.63 |
| 20150831 | 0.68 | 20190609 | 0.81 |
| 20160430 | 0.82 | 20190612 | 1.00 |
| 20160903 | 0.63 | 20200522 | 1.00 |
| 20161021 | 0.28 | 20200607 | 1.00 |

**Comment 37:**- Figures 12 and 14: what appears clearly here is the advantage given to the LR model that has been calibrated. I think this reflects the limits of the chosen methodology. It would have been better to split the dataset in calibration/validation, and to compare the calibrated versions of the three models

Thank you for your suggestions. We have split the dataset in calibration/validation, and compared the calibrated results of the three models (See comment 31).

**Comment 38:**- Table 6 It should be mentioned more clearly here what is meant by "current theory"

Thank you for your comments. Current theory means the TDUH method, and we will revise it in the revised version.

**Comment 39:**- l.512-522 and figure 15: this development should be found in the methodology section since it illustrates the computation of DTDUHs

Thank you for your comments. l.512-522 and Fig. 15 will be included in the methodology.

**Comment 40:**- l.562-565 Unfortunately, these results are not shown (appendix not present)

Thank you for your comments. We have added the results of XAJ+LR, XAJ+TDUH, and XAJ+ DTDUH models, which were calibrated separately.

**Comment 41:**- l.566-568 I think keeping the same parameters set for the runoff generation module (calibrated with another routing scheme) is not adapted here, since it corresponds more or less to an absence of calibration. This choice may advantage one of the both routing schemes. Providing calibration / validation results for both models (together with a

comparison of calibrated parameters sets), would be more relevant in my opinion.

Thank you for your suggestions. We have calibrated the XAJ+LR, XAJ+TDUH, and XAJ+ DTDUH models, separately, and these results will be added in the revised version. The added results are shown in Comment 31. In addition, the calibrated parameter sets for the two basins are given in Table 6, and comparisons will also be added in the revised version.

**Table 6**. Calibrated parameters of the three runoff routing methods for the Longhu and Dongshi River basins

| Parameters | Longhu | | | Dongshi | | |
|---|---|---|---|---|---|---|
| | LR | TDUH | DTDUH | LR | TDUH | DTDUH |
| *UM* | 9.65 | 7.13 | 8.29 | 5.38 | 8.16 | 9.13 |
| *LM* | 86.32 | 85.97 | 81.23 | 85.94 | 66.54 | 85.21 |
| *DM* | 43.96 | 47.26 | 49.25 | 47.14 | 28.53 | 45.65 |
| *B* | 0.13 | 0.39 | 0.36 | 0.40 | 0.40 | 0.40 |
| *IM* | 0.09 | 0.10 | 0.10 | 0.26 | 0.02 | 0.20 |
| *KC* | 0.12 | 0.80 | 0.80 | 1.48 | 1.50 | 1.44 |
| *C* | 0.12 | 0.12 | 0.12 | 0.16 | 0.15 | 0.12 |
| *SM* | 23.93 | 33.84 | 35.98 | 50.00 | 50.00 | 50.00 |
| *EX* | 1.19 | 1.24 | 1.10 | 1.00 | 1.00 | 1.00 |
| *KI* | 0.63 | 0.43 | 0.41 | 0.17 | 0.11 | 0.13 |
| *KG* | 0.07 | 0.27 | 0.29 | 0.53 | 0.59 | 0.57 |
| *CI* | 0.20 | 0.51 | 0.56 | 0.51 | 0.52 | 0.49 |
| *CG* | 0.94 | 0.95 | 0.94 | 0.99 | 0.99 | 0.99 |
| *CS* | 0.99 | - | - | 1.00 | - | - |
| *L* | 0.00 | 0.00 | 0.00 | 0.00 | 0.00 | 0.00 |

---

## Author Comment (AC2)

Dear Pro. Dr. Ilhan,

The authors would like to thank you for your time and constructive comments. Our point-to-point responses are listed below, where our responses are in blue, and the reviewers' comments are in black.

Kind regards, all authors

**General comments**

Yi et al. propose a novel method for computing unit hydrographs that can better account for spatial heterogeneity. For this, they relate the unit hydrograph to the dynamics of the saturated area inside the catchment, computed through the topographic wetness index for unit precipitation. The exploration of unit hydrographs and hydrological response units is an important research direction in hydrological modelling. Here, especially the impact of spatial heterogeneity is---in my opinion---underexplored. The topic of the paper is thus timely and relevant to the readership of this journal.

My main issue is with the presentation of the results and subsequent discussion. I hope that this is not nitpicking, but the current organization of the manuscript made reading, understanding, and evaluating the novelty and the methodology difficult to me. I therefore suggest revision as discussed below. Because the revision might become substantial, I recommend major revision of the current manuscript.

**Response:**

Thank you for your constructive comments, we will rewrite the discussion based on your comments. The revised discussion will include two parts:

1) Section 4.3 will be moved into the discussion part. In this section, we focus on "Errors due to spatial scale mismatch between runoff generation and runoff routing", and we will add the reasons for the potential improvements presented in the two real case studies;

2) The second part, advantages and limitations. In this section, we will summarize the advantages of the proposed method over the TDUH, as well as the potential limitations. We will make the connection to the test cases more obvious. In addition, relevant literature will be added in this Section.

**Minor comments**

**Comment 1:**L11: "... challenging hydrological modelling" reads a bit awkward. Rethink or remove "challenging."

Thank you for your comments. It will be revised as ": The spatial scale mismatch between runoff generation and runoff routing is an acceptable compromise but a common issue in hydrological modelling."

**Comment 2:**Table 6: The phrase "A typical hydrograph of direct runoff which gets generated from one centimeter of effective rainfall falling at a uniform rate over the saturated drainage basin uniformly during a specific duration." is not very clear to me. Does "uniformly" mean "spatially uniform" as implied in the Assumptions?

Sorry for being unclear here. It means spatially uniform here. To ensure the accuracy of the definition, we have read the relevant literature carefully. The unit hydrograph of a watershed is defined as a direct runoff hydrograph that results from 10 mm of excess rainfall that is generated uniformly over the drainage area at a constant rate for an effective duration (Sherman, 1932). Straub et. al. (2000) defined unit hydrograph as a discharge time graph (hydrograph) of a unit volume of direct runoff resulting from a spatially uniform distributed effective precipitation with a uniform intensity over a given duration. Bedient and Huber (2002) defined unit hydrograph as basin outflow resulting from 1.0 inch of direct runoff generated uniformly over the drainage area at a uniform rainfall rate during a specific period of rainfall duration. DTDUH is similar to the definition above, and the differences lie in that DTDUH is computed based on the generating area instead of the whole basin. However, its assumptions remained unchanged as the traditional unit hydrograph, such as a spatially uniform distributed effective precipitation.

**References:**

Bedient, B. P., Huber, C. W., 2002. Hydrology and Floodplain Analysis. Prentice-Hall, Upper Saddle River, United States of America.

Sherman, L. K, 1932. Streamflow from rainfall by the unit-graph method, Eng. News-Rec., 108, 501–505.

Straub, D. T., Melching, S. C., Kocher, E. K., 2000. Equations for Estimating Clark Unit-Hydrograph Parameters for Small Rural Watersheds in Illinois. U.S Department of the Interior U.S Geological Survey, Water- Resources Investigations Report 00-4184.

**Comment 3:** It may always be debatable whether a certain part goes into the Discussion or somewhere else. Nevertheless, I have some suggestions. In my opinion, the Discussion section should focus on discussing the presented data and results and

perhaps generalize some key insights if the data permits. The authors do this, for example, in L562-565 in the Conclusions. Based on this reasoning, I suggest a thorough rewrite of the discussion in this paper. The aim of this rewrite should be to support the discussed points with data generated in the test cases.

Thank you for your constructive comments, we will rewrite the discussion based on your comments. The revised discussion will include two parts:

1) Section 4.3 will be moved into the discussion part. In this section, we focus on "Errors due to spatial scale mismatch between runoff generation and runoff routing", and we will add the reasons for the potential improvements presented in the two real case studies;

2) The second part, advantages and limitations. In this section, we will summarize the advantages of the proposed method over the TDUH, as well as the potential limitations. We will make the connection to the test cases more obvious. In addition, relevant literature will be added in this Section.

Specifically, I have the following comments:

**Comment 4:**L498: The title "Forecasting performance advantage analysis of the proposed DTDUH" sounds a bit strange. I think it is the advantage over the TDUH algorithm? If this is the case, it should be in the title. But the advantage is not very clear to me. It is related to how the water is redistributed and how it connects to the unit hydrograph, but there is no indication what the reference hydrograph should look like.

Thank you for your suggestions. This section will be integrated into Section 5.2 "Advantages and limitations", as we cannot provide the reference hydrograph, we will add some relevant literature to explain the rationality of the proposed method.

**Comment 5:**L501-509 including Table 6: This reads as introductory information rather than a discussion. Perhaps move it into the Introduction of the paper or make the connection to the test cases more obvious to the reader.

Thank you for your comments. Table 6 will be integrated into Section 5.2 "Advantages and limitations", and we will make the connection of Table 6 to the test cases more obvious. The revised sentences are as follows:

"There are many reasons why DTDUHs simulation is superior than others, and we summarized the main differences of the TDUH and DTDUHs, including their definition and assumptions. The DTDUH was defined as a typical hydrograph of direct runoff which gets generated from one centimetre of effective rainfall falling at a uniform rate over the

saturated drainage basin uniformly during a specific duration. This realization was significant different with the understandings from Sherman (1932), who defined the unit hydrograph of a watershed as a direct runoff hydrograph that results from 10 mm of excess rainfall that is generated uniformly over the drainage area at a constant rate for an effective duration. The proposed DTDUH was computed based on the runoff generation areas instead of the whole basin, and this is the main advantages of DTDUH over TDUHs. Simultaneously, the assumption of the DTDUH remained unchanged as the traditional unit hydrograph, such as a spatially uniform distributed effective precipitation. Some researches also did similar research. For example, Andrieu et al. (2021) proposed an Event-specific Geomorphological Instantaneous Unit Hydrograph (E-GIUH), and the method relies on the width function-based GIUH (Rigon et al., 2016), as adapted to take into account the spatial variability of rainfall through replacing the width function by the rainfall width function."

**Comment 6:** L512-530 including Figure 15: This entire discussion is disconnected from the test cases. I have no doubt the discussion is valid, but I wonder if this could be better connected to the test cases. Instead of the idealized basin with 24 cells, can you show and discuss these effects in the test cases you have shown? This would be the best option. Otherwise, include a modelling test case with the idealized basin at the beginning of your test cases and then discuss it in the Discussion section where you show these effects.

Thank you for your comments. We will move Fig. 15 into methodology, and Section 5.1 in the original manuscript will be replaced by Section 4.3, and we think it will be more relevant here.

**Comment 7:** L531-549: This section is also not supported by data shown. These generalized shortcomings seem to belong in the conclusions. I suggest discussing these limitations in the context of the test cases you show. For example, if the unit hydrograph in your test case deviates from the reference hydrograph, you may show that this is most likely due to hybrid runoff generation. This would support the discussed points in this section.

Thank you for your comments. We will discuss these limitations in the context of the test cases in more detail. The revised Section 5.2 is as follows:

**5.2 Advantages and limitations of the proposed DTDUH**

We found that the accuracy of DTDUHs varies in different basins, specifically, performances of the DTDUH in Longhu Basin are more similar to that of the TDUHs, while the simulation results in Dongshi Basin are more consistent with that of the LR method. In general, the DTDUHs performed the best over the three runoff routing methods

for both test cases. There are many reasons why DTDUHs simulation is superior than others, and we summarized the main differences of the TDUH and DTDUHs, including their definition and assumptions. The DTDUH was defined as a typical hydrograph of direct runoff which gets generated from one centimetre of effective rainfall falling at a uniform rate over the saturated drainage basin uniformly during a specific duration. This realization was significant different with the understandings from Sherman (1932), who defined the unit hydrograph of a watershed as a direct runoff hydrograph that results from 10 mm of excess rainfall that is generated uniformly over the drainage area at a constant rate for an effective duration. The proposed DTDUH was computed based on the runoff generation areas instead of the whole basin, and this is the main advantages of DTDUH over TDUHs. Simultaneously, the assumption of the DTDUH remained unchanged as the traditional unit hydrograph, such as a spatially uniform distributed effective precipitation. Some researches also did similar research. For example, Andrieu et al. (2021) proposed an Event-specific Geomorphological Instantaneous Unit Hydrograph (E-GIUH), and the method relies on the width function-based GIUH (Rigon et al., 2016), as adapted to take into account the spatial variability of rainfall through replacing the width function by the rainfall width function.

[Figure]

**Figure 3.** Details of the runoff components of the 16 flood events for the (a) Longhu River basin. (b) Dongshi River basin.

Although DTDUH showed advantages in both basins, the degree of improvement compared with TDUH was not consistent. Therefore, we summarized the potential limitations of the DTDUH. First, we utilized the DTDUH only for the surface runoff in the both basins, proportions of the subsurface storm flow and subsurface runoff may cause considerable interference to the simulation results. Runoff components of the Longhu and Dongshi River basins are as given in Fig. 1. Results shown that the average surface runoff ratio in Longhu Basin and Dongshi basin is 64.4% and 80.3%, respectively. This result suggested that the improvement in accuracy caused by DTDUH may be more significant in the Dongshi basin. And, this conclusion is consistent with the calibration and verification results. Second, a hybrid runoff generation process pattern formed by more than one

mechanism can often be identified in semi-humid, semi-arid and mountain watershed, because of the heterogeneity of underlying surface conditions and meteorological factors (Hu et al., 2021; Yi et al., 2023). When there occurs more than the saturation-excess rainfall, the saturated area extraction method based on the TWI will not be applicable as the excess rainfall can also be generated from the unsaturated areas. Fig. 2 shows the antecedent soil moisture conditions for the Longhu and Dongshi River basins. It can be found that the antecedent soil moisture can be low for some flood events, such as No. 19790607, 19890502, 20150509 and so on. When the antecedent soil moisture is low and the rainfall intensity is high, the drainage basin may produce not only the saturation excess, which results in low accuracy of the DTDUH method.

[Figure]

**Figure 4.** Details of the antecedent soil moisture of the 16 flood events for the (a) Longhu River basin. (b) Dongshi River basin.

**References:**

Andrieu, H., Moussa, R., Kirstetter, P.-E., 2021. The Event-specific Geomorphological Instantaneous Unit Hydrograph (E-GIUH): The basin hydrological response characteristic of a flood event. Journal of Hydrology, 603: 127158. https://doi.org/10.1016/j.jhydrol.2021.127158

Hu, C. H., Ran, G., Li, G., Yu, Y., Wu, Q., Yan, D. H., Jian, S. Q., 2021. The effects of rainfall characteristics and land use and cover change on runoff in the Yellow River basin, China. Journal of Hydrology and Hydromechanics, 69(1): 29-40. https://doi.org/10.2478/johh-2020-0042

Yi, B., Chen, L., Liu, Y., Guo, H., Leng, Z., Gan, X., Xie, T., Mei, Z., 2023. Hydrological modelling with an improved flexible hybrid runoff generation strategy. Journal of Hydrology, 620: 129457. https://doi.org/10.1016/j.jhydrol.2023.129457

Rigon, R., Bancheri, M., Formetta, G. and de Lavenne, A., 2016. The geomorphological unit hydrograph from a historical-critical perspective. Earth Surface Processes and

Landforms, 41(1), 27-37. https://doi.org/10.1002/esp.3855

Sherman, L. K, 1932. Streamflow from rainfall by the unit-graph method, Eng. News-Rec., 108, 501–505.